# Multistep loading of a DNA sliding clamp onto DNA by replication factor C

**Marina Schrecker[1†], Juan C Castaneda[2,3†], Sujan Devbhandari[3], Charanya Kumar[3], Dirk Remus[3]\*, Richard K Hite[1]\***

[1]Structural Biology Program, Memorial Sloan Kettering Cancer Center, New York, United States; [2]Weill Cornell Medicine Graduate School, Weill Cornell Medicine, New York, United States; [3]Molecular Biology Program, Memorial Sloan Kettering Cancer Center, New York, United States

**Abstract:** The DNA sliding clamp proliferating cell nuclear antigen (PCNA) is an essential co-factor for many eukaryotic DNA metabolic enzymes. PCNA is loaded around DNA by the ATP-dependent clamp loader replication factor C (RFC), which acts at single-stranded (ss)/double-stranded DNA (dsDNA) junctions harboring a recessed 3′ end (3′ ss/dsDNA junctions) and at DNA nicks. To illuminate the loading mechanism we have investigated the structure of RFC:PCNA bound to ATPγS and 3′ ss/dsDNA junctions or nicked DNA using cryogenic electron microscopy. Unexpectedly, we observe open and closed PCNA conformations in the RFC:PCNA:DNA complex, revealing that PCNA can adopt an open, planar conformation that allows direct insertion of dsDNA, and raising the question of whether PCNA ring closure is mechanistically coupled to ATP hydrolysis. By resolving multiple DNA-bound states of RFC:PCNA we observe that partial melting facilitates lateral insertion into the central channel formed by RFC:PCNA. We also resolve the Rfc1 N-terminal domain and demonstrate that its single BRCT domain participates in coordinating DNA prior to insertion into the central RFC channel, which promotes PCNA loading on the lagging strand of replication forks in vitro. Combined, our data suggest a comprehensive and fundamentally revised model for the RFC-catalyzed loading of PCNA onto DNA.

**\*For correspondence:**
hiter@mskcc.org (RKH);
remusd@mskcc.org (DR)

[†]These authors contributed equally to this work

## Editor's evaluation

The present work uses structural approaches to describe how an ATPase known as a 'clamp loader' opens a ring-shaped clamp protein and binds DNA to promote the deposition of the clamp around a nucleic acid duplex to support chromosomal replication. The paper is important in that it reports new findings on how different regions of the clamp loader bind to and open a clamp, and how the enzyme engages single-stranded and double-stranded regions of target DNAs. Different conformational states of the clamp loader and the clamp are observed, providing a molecular picture of several steps in the clamp loading cycle.

## Introduction

Proliferating cell nuclear antigen (PCNA) is an essential co-factor in DNA metabolic processes critical for the maintenance of eukaryotic chromosomes. Initially characterized as a processivity factor for replicative DNA polymerases, PCNA has since been shown to be an interaction hub for a wide array of proteins involved in DNA replication, the repair or bypass of DNA damage, chromatin assembly, chromosome cohesion, and cell cycle regulation (*Boehm et al., 2016*). Integral to PCNA function is its ability to topologically encircle DNA and act as a sliding clamp. PCNA forms homotrimeric complexes in which the subunits are arranged head-to-tail to form a closed ring (*Gulbis et al., 1996*; *Krishna*

*et al., 1994*). The outer ring surface is shaped by globular β-sheet domains and harbors the protein interaction sites, while the ring interior is lined with α-helices featuring positively charged surface residues that can engage in electrostatic interactions with the DNA phosphate backbone. With an inner diameter of ~34 Å the PCNA ring can comfortably accommodate B-form DNA (~20 Å diameter), facilitating the ability of PCNA to slide along DNA (*Li et al., 2021*).

Importantly, PCNA also forms stable rings free in solution, preventing its spontaneous binding to DNA in the absence of free DNA ends (*Binder et al., 2014*; *Yao et al., 1996*; *Zhuang et al., 2006*). Consequently, PCNA loading onto DNA is facilitated by clamp loader complexes belonging to the AAA+ family of ATPases that open the PCNA ring and close it around DNA (*Kelch, 2016*). Two related clamp loaders, RFC and Ctf18-RFC, can load PCNA onto DNA. RFC is essential for cell growth and considered the canonical PCNA loader, while Ctf18-RFC is non-essential and appears to perform more specialized roles in replication checkpoint signaling and chromosome cohesion (*Arbel et al., 2021*). The DNA substrate specificity of clamp loaders is critical to prevent random and futile clamp loading along chromosome arms and target clamps to relevant sites instead. Accordingly, RFC loads PCNA specifically at single-stranded (ss)/double-stranded DNA (dsDNA) junctions harboring a recessed 3' end (3' junctions) or at DNA nicks, DNA structures that are commonly formed at sites of DNA replication and repair (*Bylund and Burgers, 2005*; *Cai et al., 1996*; *Ellison and Stillman, 2003*; *Hayner et al., 2014*; *Lee et al., 1991*; *Tsurimoto and Stillman, 1991*). However, how RFC recognizes 3' junctions, opens the PCNA ring, and inserts DNA into the PCNA ring remains unclear due to a lack of structural information on the RFC:PCNA:DNA complex.

The overall structural organization of RFC has been revealed by X-ray crystallography and cryo-EM analyses of the yeast and human complexes, respectively (*Bowman et al., 2004*; *Gaubitz et al., 2020*). RFC is composed of four small AAA+ subunits, Rfc2-5, and one large AAA+ subunit, Rfc1, which assemble head-to-tail into a two-tiered partially open ring structure. The N-terminal tier contains the AAA+ domains and the unique A' domain of Rfc1 (note: yeast nomenclature is used throughout this paper). The C-terminal tier, in contrast, forms a rigid collar composed of α-helical domains provided by Rfc1-5. As is characteristic for AAA+ ATPases, ATP is bound at RFC subunit interfaces with catalytic residues at each ATPase site contributed by adjacent protomers (*Johnson et al., 2006*). However, Rfc5 lacks residues critical for ATP hydrolysis and does not form a composite active site due to its position at the gap in the AAA+ tier. Mutational studies suggest that all four ATPase sites in RFC are required for normal RFC function (*Johnson et al., 2006*; *Sakato et al., 2012a*; *Schmidt et al., 2001a*; *Schmidt et al., 2001b*; *Marzahn et al., 2014*; *Cai et al., 1998*; *Podust et al., 1998a*). Biochemical studies moreover suggest that ATP-binding to RFC promotes sequential PCNA binding, PCNA opening, and DNA binding by the RFC:PCNA complex, while ATP hydrolysis induces the release of closed PCNA on DNA (*Zhuang et al., 2006*; *Chen et al., 2009*; *Gomes et al., 2001b*; *Sakato et al., 2012b*). The structural basis for these ATP-driven events is not clear.

In previous structural studies, which lacked DNA, RFC adopts an inactive or autoinhibited conformation in complex with a closed PCNA ring in which the ATPase sites are incompletely formed (*Bowman et al., 2004*; *Gaubitz et al., 2020*). In these structures, PCNA contacts Rfc1, -4, and -3, but is freely suspended over Rfc2 and -5. However, PCNA clamp opening is thought to involve interactions with all RFC subunits, which was suggested to force the closed planar PCNA ring into an open spiral conformation that matches the pitch of the spirally arranged RFC subunits (*Bowman et al., 2004*). Such an out-of-plane clamp-opening mechanism is supported by molecular dynamics (MD) simulations of the PCNA ring and structural analyses of homologous clamp loader:clamp:DNA complexes from bacteriophage T4 and archaea (*Kazmirski et al., 2005*; *Kelch et al., 2011*; *Miyata et al., 2005*). In this model, PCNA will form a gap of <10 Å, which would preclude the insertion of dsDNA. However, by analogy to the clamp loader complexes from *Escherichia coli* and bacteriophage T4, and in accordance with previous DNase footprinting and mutational analyses, RFC is thought to bind to the duplex region of 3' junctions inside its central channel, loading PCNA onto the adjacent dsDNA (*Tsurimoto and Stillman, 1991*; *Kelch et al., 2011*; *Simonetta et al., 2009*; *Yao et al., 2006*). A 'screw-cap model' that reconciles the narrow gap predicted for the open RFC-PCNA complex with PCNA loading over dsDNA was proposed in which the open RFC-PCNA complex initially threads onto ssDNA and subsequently slides toward the 3' junction for functional positioning (*Bowman et al., 2004*). Direct evidence for this model is currently lacking.

In addition to the AAA+ and collar domains, Rfc1 contains unique N- and C-terminal extensions that are absent in Rfc2-5. The C-terminus of Rfc1 contains the A' domain that is situated between the AAA+ domains of Rfc1 and Rfc5 (*Bowman et al., 2004*; *Gaubitz et al., 2020*). The structural organization of the Rfc1 N-terminal domain (NTD) in the context of the RFC complex is not known. It has been noted early on that a region of the Rfc1 NTD exhibits significant homology to domains found in prokaryotic DNA ligases and PARP1 (*Bunz et al., 2006*; *Burbelo et al., 2006*; *Fotedar et al., 1996*). Subsequent sequence analysis suggested that the Rfc1 NTD encompasses a single BRCT domain that forms a distinct BRCT subclass with single BRCT domains found in prokaryotic DNA ligases and PARP1 (*Bork et al., 1997*). This prediction was confirmed by NMR analysis of the human Rfc1 BRCT domain (*Kobayashi et al., 1997a*; *Kobayashi et al., 1997b*). Surprisingly, unlike tandem BRCT domains in eukaryotes that mediate phosphorylation-dependent protein interactions, the BRCT domain of Rfc1 was found to harbor a structure-specific DNA-binding activity that has increased affinity for dsDNA featuring a recessed and phosphorylated 5' end (*Burbelo et al., 2006*; *Fotedar et al., 1996*; *Kobayashi et al., 1997b*; *Allen, 1997*).

The functional significance of the Rfc1 NTD has remained enigmatic. On the one hand, biochemical studies with truncated RFC complexes have demonstrated that the Rfc1 NTD is not essential for PCNA loading in vitro or for normal growth of budding yeast cells (*Gomes et al., 1997*; *Uhlmann et al., 1997*; *Podust et al., 1998b*). Moreover, RFC lacking the Rfc1 NTD was found to exhibit even greater activity in vitro than the full-length protein, which has been variably attributed to the greater protein stability or the loss of non-specific DNA-binding activity of RFC lacking the Rfc1 NTD (*Gomes et al., 1997*; *Uhlmann et al., 1997*; *Podust et al., 1998b*). On the other hand, mutations in the Rfc1 NTD cause a cold-sensitive growth defect in budding yeast, while both truncation and mutation of the Rfc1 NTD increase sensitivity of budding yeast cells to the DNA-damaging agent methyl methanesulfonate (MMS) (*Gomes et al., 1997*; *McAlear et al., 1996*; *Xie et al., 1999*). Truncation of the NTD in human RFC1 has also been associated with Hutchinson-Gilford progeria syndrome (*Tang et al., 2012*). Combined with the high degree of conservation of the Rfc1 NTD across eukaryotes, these data suggest that the Rfc1 NTD performs an important but undefined function (*Cullmann et al., 1995*).

To address the mechanism of PCNA loading by RFC, we have reconstituted the budding yeast RFC:PCNA complex bound to a DNA substrate harboring a recessed 3' junction and examined its structure using cryogenic electron microscopy. Contrary to previous models, our data reveal that PCNA is opened in a largely planar fashion and that PCNA closing around DNA can occur in the absence of ATP hydrolysis. Moreover, by resolving multiple DNA-bound states of the RFC:PCNA complex, we uncover the mechanism by which dsDNA is inserted into the central channel formed by RFC:PCNA. Finally, we resolve the DNA-bound structure of the N-terminal BRCT domain in the context of RFC:PCNA:DNA complexes and demonstrate that this domain promotes PCNA loading at replication forks in vitro. Combined, our data suggest a novel and comprehensive model for the RFC-catalyzed loading of PCNA at 3' junctions that refutes previous models.

## Results

### Structure of RFC:PCNA in an active state

We expressed and purified the full-length RFC complex composed of Rfc1-5 from *Saccharomyces cerevisiae* cells, and trimeric PCNA and the replication factor A (RPA) complex from *E. coli* (*Figure 1—figure supplement 1*). The ATPase activity of purified RFC was stimulated by the presence of PCNA and DNA containing a 3' ss/dsDNA junction, indicating that the purified components are functional. RPA has been shown to promote the loading of PCNA by RFC specifically at 3' ss/dsDNA junctions (*Hayner et al., 2014*). Therefore, to resolve the structure of RFC actively loading PCNA onto DNA, we preincubated RPA with a DNA substrate containing a 20-base double-stranded segment and a 50-base 5' overhang, then assembled the full RFC:PCNA:DNA complex in the presence of saturating ATPγS, separated intact complexes by glycerol gradient centrifugation, and collected cryogenic electron microscopic images (*Figure 1—figure supplement 1* and *Figure 1—figure supplement 2*). Image analysis revealed the presence of multiple distinct conformations of RFC:PCNA in the presence of a 3' ss/dsDNA junction at resolutions from 2.9 to 2.1 Å (*Figure 1—figure supplement 2* and *Table 1*). In all of the conformations, RFC:PCNA is composed of three layers: a bottom layer comprised of the collar

**Table 1.** Cryo-EM data collection, refinement and validation statistics.

| | RFC-PCNA DNA1 open (EMD-27663) (PDB 8DQX) | RFC-PCNA DNA1 intermediate (EMD-27666) (PDB 8DQZ) | RFC-PCNA DNA1 closed (EMD-27667) (PDB 8DR0) | RFC-PCNA DNA2 closed consensus (EMD-27668) (PDB 8DR1) | RFC-PCNA DNA2 closed with NTD (EMD-27669) (PDB 8DR3) | RFC-PCNA DNA2 open no NTD (EMD-27670) (PDB 8DR4) | RFC-PCNA DNA2 open with NTD (EMD-27671) (PDB 8DR5) | RFC-PCNA nicked DNA closed (EMDB-27672) (PDB 8DR6) | RFC-PCNA nicked DNA open (EMDB-27673) (PDB 8DR7) | Rad24-RFC Open (EMD-27662) (PDB 8DQW) |
|---|---|---|---|---|---|---|---|---|---|---|
| **Data collection and processing** | | | | | | | | | | |
| Magnification | 29,000 × | 29,000 × | 29,000 × | 29,000 × | 29,000 × | 29,000 × | 29,000 × | 29,000 × | 29,000 × | 29,000 × |
| Voltage (kV) | 300 keV | 300 keV | 300 keV | 300 keV | 300 keV | 300 keV | 300 keV | 300 keV | 300 keV | 300 keV |
| Electron exposure (e–/Å$^2$) | 66 | 66 | 66 | 66 | 66 | 66 | 66 | 66 | 66 | 66 |
| Defocus range (μm) | –0.5 to –2.0 | –0.5 to –2.0 | –0.5 to –2.0 | –0.5 to –2.0 | –0.5 to –2.0 | –0.5 to –2.0 | –0.5 to –2.0 | –0.5 to –2.0 | –0.5 to –2.0 | –0.5 to –2.0 |
| Pixel size (Å) | 0.826 | 0.826 | 0.826 | 0.826 | 0.826 | 0.826 | 0.826 | 0.826 | 0.826 | 0.826 |
| Symmetry imposed | C1 | C1 | C1 | C1 | C1 | C1 | C1 | C1 | C1 | C1 |
| Initial particle images (no.) | 5,688,448 | 5,688,448 | 5,688,448 | 5,688,448 | 3,356,580 | 3,356,580 | 3,356,580 | 2,031,079 | 2,031,079 | 4,117,022 |
| Final particle images (no.) | 616,330 | 41,190 | 252,647 | 872,447 | 356,424 | 104,742 | 61,483 | 359,126 | 142,294 | 938,420 |
| Map resolution (Å) | 2.1 | 2.92 | 2.42 | 2.14 | 2.2 | 2.45 | 2.76 | 2.39 | 2.7 | 2.1 |
| FSC threshold | 0.143 | 0.143 | 0.143 | 0.143 | 0.143 | 0.143 | 0.143 | 0.143 | 0.143 | 0.143 |
| **Refinement** | | | | | | | | | | |
| Initial model used (PDB code) | Closed state | Open state | 1SXJ | Closed state | Closed state | Open state | Open state | Closed state | Open state | 7ST9 |

*Table 1 continued on next page*

*Table 1 continued*

| | RFC-PCNA DNA1 open (EMD-27663) (PDB 8DQX) | RFC-PCNA DNA1 intermediate (EMD-27666) (PDB 8DQZ) | RFC-PCNA DNA1 closed (EMD-27667) (PDB 8DR0) | RFC-PCNA DNA2 closed consensus (EMD-27668) (PDB 8DR1) | RFC-PCNA DNA2 closed with NTD (EMD-27669) (PDB 8DR3) | RFC-PCNA DNA2 open no NTD (EMD-27670) (PDB 8DR4) | RFC-PCNA DNA2 open with NTD (EMD-27671) (PDB 8DR5) | RFC-PCNA nicked DNA closed (EMDB-27672) (PDB 8DR6) | RFC-PCNA nicked DNA open (EMDB-27673) (PDB 8DR7) | Rad24-RFC Open (EMD-27662) (PDB 8DQW) |
|---|---|---|---|---|---|---|---|---|---|---|
| Model resolution (Å) | 1.99 | 2.87 | 2.25 | 2.05 | 2.13 | 2.3 | 2.69 | 2.3 | 2.66 | 2.12 |
| FSC threshold | 0.5 | 0.5 | 0.5 | 0.5 | 0.5 | 0.5 | 0.5 | 0.5 | 0.5 | 0.5 |
| Model composition | | | | | | | | | | |
| Non-hydrogen atoms | 22,030 | 21,451 | 21,514 | 22,004 | 23,129 | 21,999 | 22,780 | 217,998 | 21,655 | 23,629 |
| Protein residues | 2,598 | 2,599 | 2,608 | 2,609 | 2,758 | 2,608 | 2,749 | 2,608 | 2,600 | 2,802 |
| Nucleotide residues | 26 | 40 | 40 | 63 | 63 | 63 | 49 | 63 | 49 | 35 |
| Ligands | 9 | 9 | 9 | 9 | 9 | 9 | 9 | 9 | 9 | 9 |
| B factors (Å²) | | | | | | | | | | |
| Protein (mean) | 10 | 75.3 | 34.7 | 10.8 | 21.5 | 50.6 | 42.5 | 50.6 | 33.9 | 9.8 |
| Nucleotide (mean) | 6.7 | 161.8 | 85 | 25.5 | 52.6 | 97.2 | 83 | 97.2 | 80.7 | 41.9 |
| Ligand (mean) | 3.2 | 56.9 | 25.2 | 5 | 11.3 | 38 | 22.9 | 39.3 | 15.5 | 7.4 |
| R.m.s. deviations | | | | | | | | | | |
| Bond lengths (Å) | 0.004 | 0.002 | 0.002 | 0.003 | 0.002 | 0.002 | 0.002 | 0.002 | 0.002 | 0.002 |
| Bond angles (°) | 0.628 | 0.515 | 0.475 | 0.528 | 0.476 | 0.479 | 0.5 | 0.479 | 0.528 | 0.479 |
| Validation | | | | | | | | | | |

*Table 1 continued on next page*

*Table 1 continued*

| | RFC-PCNA DNA1 open (EMD-27663)(PDB 8DQX) | RFC-PCNA DNA1 intermediate (EMD-27666)(PDB 8DQZ) | RFC-PCNA DNA1 closed (EMD-27667)(PDB 8DR0) | RFC-PCNA DNA2 closed consensus (EMD-27668)(PDB 8DR1) | RFC-PCNA DNA2 closed with NTD (EMD-27669)(PDB 8DR3) | RFC-PCNA DNA2 open no NTD (EMD-27670)(PDB 8DR4) | RFC-PCNA DNA2 open with NTD (EMD-27671)(PDB 8DR5) | RFC-PCNA nicked DNA closed (EMDB-27672)(PDB 8DR6) | RFC-PCNA nicked DNA open (EMDB-27673)(PDB 8DR7) | Rad24-RFC Open (EMD-27662)(PDB 8DQW) |
|---|---|---|---|---|---|---|---|---|---|---|
| MolProbity score | 1.31 | 1.3 | 1.17 | 1.12 | 1.19 | 1.12 | 1.29 | 1.12 | 1.28 | 1.15 |
| Clashscore | 5.05 | 5.53 | 3.68 | 3.25 | 3.22 | 3.25 | 4.82 | 3.25 | 5.17 | 3.61 |
| Poor rotamers (%) | 1.13 | 0.26 | 1.04 | 0.65 | 1.28 | 0 | 0.45 | 0.87 | 0.82 | 0.4 |
| Ramachandran plot | | | | | | | | | | |
| Favored (%) | 98.06 | 98.06 | 98.65 | 98.65 | 98.43 | 98.61 | 97.84 | 98.61 | 98.1 | 98.48 |
| Allowed (%) | 1.94 | 1.94 | 1.35 | 1.35 | 1.57 | 1.39 | 2.16 | 1.39 | 1.9 | 1.52 |
| Disallowed (%) | 0.00 | 0.00 | 0.00 | 0.00 | 0.00 | 0.00 | 0.00 | 0.00 | 0.00 | 0.00 |

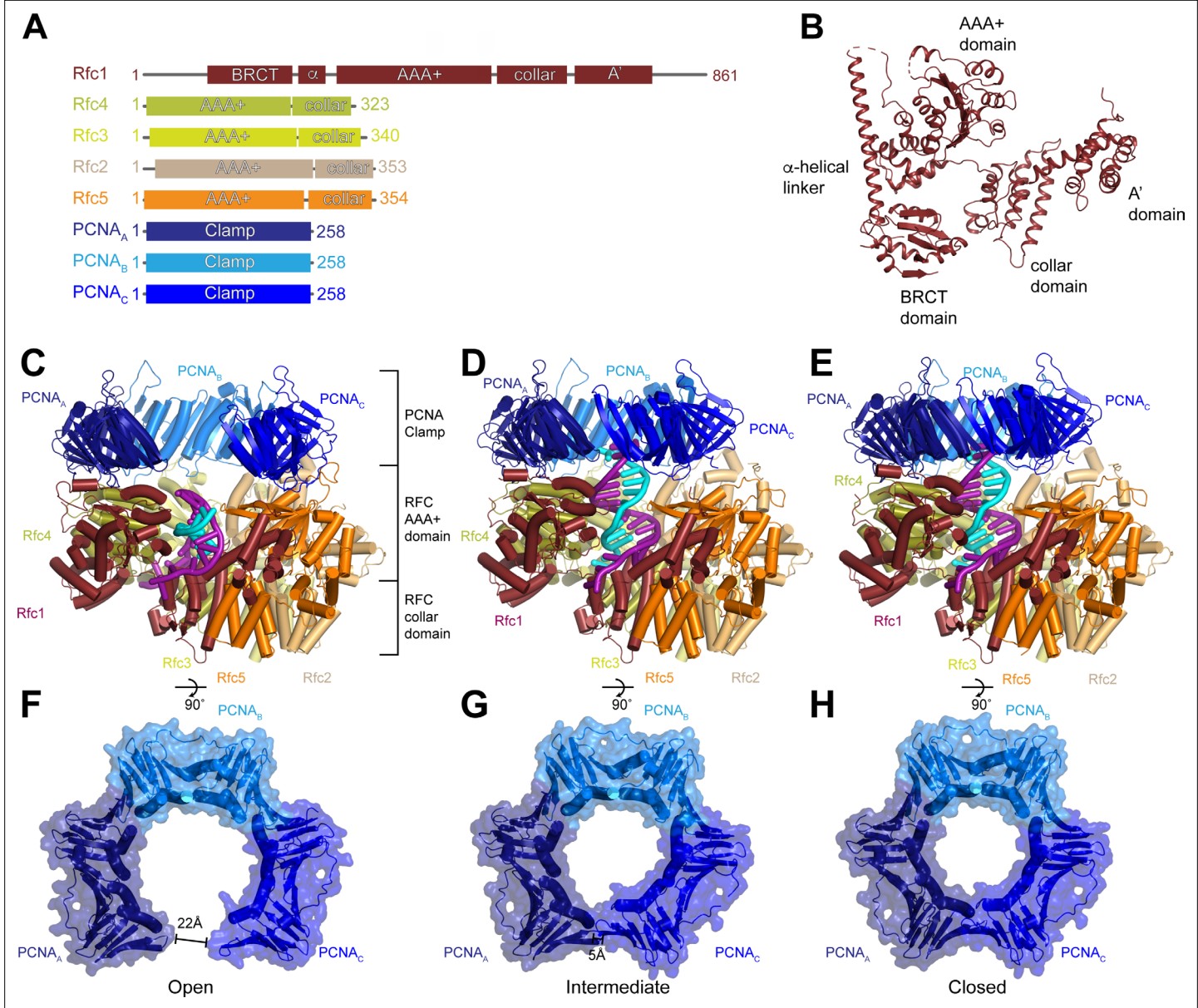

**Figure 1.** Structures of replication factor C (RFC) in active states in complex with proliferating cell nuclear antigen (PCNA). (**A**) Domain arrangement of subunits resolved in RFC-PCNA structures. (**B**) Structure of yeast Rfc1.( **C–E**) Structures of RFC:PCNA in open (**C**), intermediate (**D**), and closed (**E**) states, colored by subunit. Rfc1 is maroon, Rfc4 is gold, Rfc3 is yellow, Rfc2 is sand, Rfc5 is orange, and the PCNA protomers are blue. DNA is shown in cyan and magenta. (**F–H**) Open (**F**), intermediate (**G**), and closed (**H**) conformations of PCNA, colored by subunit.

The online version of this article includes the following video, source data, and figure supplement(s) for figure 1:

**Figure supplement 1.** Purification and analysis of RFC:PCNA.

**Figure supplement 1—source data 1.** Uncropped images of gels presented in *Figure 1—figure supplement 1*.

**Figure supplement 2.** Cryo-EM analysis of RFC:PCNA with DNA substrate 1 (DNA$_1$).

**Figure 1—video 1.** Morph between open and closed states of RFC:PCNA with DNA substrate 1.

https://elifesciences.org/articles/78253/figures#fig1video1

domains of the five RFC subunits, a middle layer of the AAA+ domains of the five RFC subunits, and the C-terminal A' domain of Rfc1 and a top layer of the PCNA homotrimer (*Figure 1A–E*). Despite being present in the vitrified sample, no unassigned densities are present in the map, indicating that RPA is not associated with either RFC or PCNA in a stable manner. The most notable difference between the three conformations is the arrangement of the PCNA ring (*Figure 1F–H* and *Figure 1—video 1*).

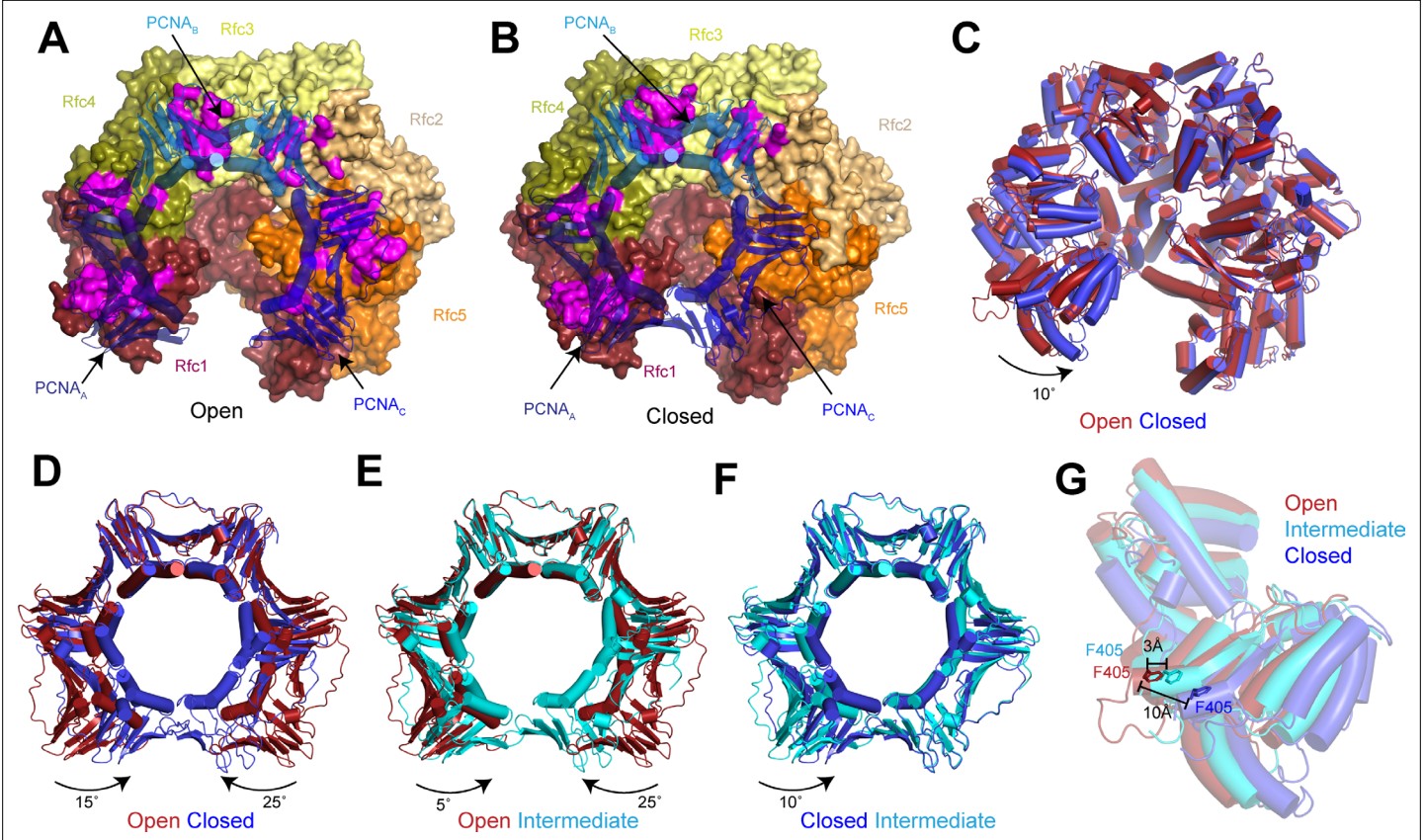

**Figure 2.** Replication factor C (RFC) stabilizes proliferating cell nuclear antigen (PCNA) in the open conformation. (**A–B**) Structures of RFC:PCNA in open (**A**) and closed (**B**) states. RFC is shown as a surface and PCNA is shown as a transparent cartoon. RFC and PCNA are colored by subunit with residues on RFC that interact with PCNA highlighted in magenta. DNA is removed for clarity. (**C**) Superposition of RFC in open (red) and closed (blue) states. Structures are aligned using Rfc3. (**D–F**) Superpositions of PCNA between open and closed (**D**), open and intermediate (**E**), and closed and intermediate states (**F**). Structures are aligned using PCNA$_B$. Open PCNA is red, intermediate PCNA is cyan, and closed PCNA is blue. (**G**) Rfc1 in open (red), intermediate (cyan), and closed (blue) states. Phe405 is shown as sticks. Structures are aligned using Rfc3.

The online version of this article includes the following video and figure supplement(s) for figure 2:

**Figure supplement 1.** Comparison of RFC:PCNA:DNA$_1$ with structures of yeast and human RFC in autoinhibited states bound to PCNA.

**Figure supplement 2.** Comparison of nucleotide densities in RFC-PCNA and Rad24-RFC.

**Figure supplement 3.** Conformational flexibility of Rfc5 α4 loop.

**Figure supplement 4.** The interface between replication factor C (RFC) and proliferating cell nuclear antigen (PCNA) is distinct in different conformations.

**Figure 2—video 1.** Morph between open and closed states of proliferating cell nuclear antigen (PCNA).

https://elifesciences.org/articles/78253/figures#fig2video1

In classes 1 and 2, PCNA is dilated in a planar fashion compared to crystal structures of closed PCNA rings (*Gulbis et al., 1996*; *Krishna et al., 1994*). In class 1, the space between PCNA protomers measures 22 Å, which is wide enough to accommodate dsDNA. The gap in class 2 is much narrower and unable to accommodate dsDNA. In class 3, PCNA is fully closed and resembles structures of PCNA alone and in complex with RFC in an inactive state (*Krishna et al., 1994*; *Bowman et al., 2004*). We will thus refer to class 1 as open, class 2 as intermediate, and class 3 as closed.

## RFC stabilizes PCNA in an open state

In the open state, the AAA+ domains of all five RFC subunits bind to PCNA (*Figure 2A*). Due to a mismatch between heteropentameric RFC and homotrimeric PCNA, there are several different types of interfaces between the subunits of RFC and PCNA. Rfc1, Rfc3, and Rfc5 bind in the hydrophobic grooves between the N- and C-terminal domains of PCNA$_A$, PCNA$_B$, and PCNA$_C$, respectively. Rfc4

and Rfc2 bind at the inter-subunit interfaces between PCNA$_A$ and PCNA$_B$, and PCNA$_B$ and PCNA$_C$, respectively. The contacts established by Rfc1, Rfc3, and Rfc5 are larger and are comprised of both hydrophobic and hydrophilic interactions, while those established by Rfc4 and Rfc2 at the inter-subunit interfaces are smaller and largely hydrophilic (*Figure 2A*). Unique among the RFC subunits, Rfc1 forms a second interaction with PCNA via its C-terminal A' domain, resulting in each PCNA protomer associating with two RFC subunits in the open state (*Figure 2A* and *Figure 2—figure supplement 4*). Together, these interactions stabilize PCNA in an open horseshoe-like shape that is patterned on the shape of the AAA+ and A' domains of RFC.

The planar orientation of PCNA relative to RFC in RFC:PCNA:DNA is distinct from the tilted orientation resolved for PCNA in an X-ray crystal structure of yeast and a cryo-EM structure human RFC:PCNA in autoinhibited, DNA-free states (*Bowman et al., 2004*; *Gaubitz et al., 2020*; *Figure 2—figure supplement 1*). Also different are the positions of the AAA+ domains of RFC. In the auto-inhibited structures, the AAA+ domains of RFC adopt a spiral configuration, whereas the AAA+ domains are arranged in a largely planar fashion in RFC:PCNA bound to a 3' ss/ds DNA junction. The rearrangement of the AAA+ domains of RFC coincides with the conserved arginine finger from the neighboring subunits moving into direct contact with the ATPγS bound to the catalytic Rfc1, Rfc4, Rfc3, and Rfc2 subunits (*Figure 2—figure supplement 1* and *Figure 2—figure supplement 2*). By contrast, GDP rather than an adenine nucleotide is bound in the nucleotide-binding site of Rfc5, which lacks residues important for ATP hydrolysis (*Bowman et al., 2004*; *Schmidt et al., 2001a*; *Cai et al., 1998*). Reprocessing of a recent data set of Rad24-RFC in complex with 9-1-1 and a 5' ss/dsDNA junction yielded a 2.1 Å reconstruction of the open state that revealed that Rfc5 also binds GDP in the Rad24-RFC complex (*Figure 2—figure supplement 2* and *Table 1*; *Castaneda et al., 2022*). In both structures, the specificity for Rfc5-binding guanine rather than adenine nucleotides is imparted by Rfc5-Arg52, which coordinates the C6 carbonyl and the N7 of the guanine base (*Figure 2—figure supplement 2*). Thus, the open state of RFC represents an active, pre-hydrolysis state of RFC with which PCNA is fully engaged.

Compared to the open state, the PCNA$_A$ and PCNA$_C$ protomers in the closed state rotate inward as rigid bodies in a planar fashion by 15° and 25°, respectively (*Figure 2D* and *Figure 2—video 1*). The inward movement of PCNA$_C$ disengages it from RFC. In absence of its interaction with PCNA, the PCNA-binding site of Rfc5 adopts an alternative conformation in which it instead binds to the N-terminus of Rfc2 (*Figure 2—figure supplement 3*). In the open state, the loop following the α4 helix, which we call the α4 loop that is comprised of residues 120–135, extends up and inserts into the hydrophobic groove of PCNA$_C$, establishing a large interface. In the closed state, the α4 loop bends down and is sandwiched between the AAA+ domain of Rfc5 and the N-terminus of Rfc2. The α4 loop of Rfc5 also undergoes a conformational change during the opening of the 9-1-1 checkpoint clamp (*Castaneda et al., 2022*). However, the conformations resolved in the presence of PCNA are distinct from those resolved in Rad24-RFC:9-1-1 as the α4 loop adopts a β-hairpin in the open Rad24-RFC:9-1-1 structure and is disordered in closed Rad24-RFC:9-1-1 (*Figure 2—figure supplement 3*). Thus, flexibility within its α4 loop allows Rfc5 to bind to two distinct DNA clamps – PCNA and 9-1-1 – and stabilize them in their open states during loading.

While the inward movement of PCNA$_C$ results in its disengagement from RFC in the closed state, PCNA$_A$ remains tightly associated with RFC despite also pivoting inward (*Figure 2B*). The interface between PCNA$_A$ and Rfc1 is unchanged in the closed state because Rfc1 also rotates compared to the open state (*Figure 2C*). The inward rotation of Rfc1 yields a 10 Å movement of Rfc1-Phe405, which is inserted into the hydrophobic groove of PCNA$_A$ in both the open and closed states (*Figure 2G*). Indeed, Rfc1-Phe405 is inserted into the central groove of all known structures of RFC in complex with PCNA, indicating that it plays a central role in stabilizing the RFC:PCNA interaction (*Figure 2—figure supplement 4*).

In the intermediate state, PCNA adopts a conformation that is a mixture of protomers that adopt open-like and closed-like states. In the intermediate state, PCNA$_C$ disengages from RFC to adopt a closed-like conformation. In contrast, Rfc1 and PCNA$_A$, which move together as a single unit in all of the structures, are only slightly rotated and adopt more open-like positions that prevent PCNA$_A$ from binding to PCNA$_C$ and closing of the ring (*Figure 2E and F* and *Figure 2—figure supplement 4*). Notably, the conformational changes in RFC in the closed and intermediate states do not arise from changes in nucleotide-binding state. Densities corresponding to Mg$^{2+}$-coordinated ATPγS are resolved

in the nucleotide-binding sites of Rfc1, Rfc4, Rfc3, and Rfc2 in the closed and intermediate maps as is a GDP in the nucleotide-binding site of the non-catalytic Rfc5 (*Figure 2—figure supplement 1* and *Figure 2—figure supplement 2*). Based on comparisons between the open, intermediate, and closed states, we propose a two-step mechanism for the opening of PCNA when complexed with ATP-bound RFC. In the first step, Rfc1 and PCNA$_A$ together pivot outward, separating PCNA$_A$ from PCNA$_C$ to form an intermediate state. In the absence of its interaction with PCNA$_A$, PCNA$_C$ can rotate freely and sample a range of conformations, including the open conformation where it binds to Rfc5. Once PCNA$_C$ contacts with Rfc5, the flexible α4 loop becomes ordered and binds in the hydrophobic groove of PCNA$_C$, stabilizing PCNA in the open state.

## 3' ss/dsDNA junctions bind to RFC:PCNA at multiple sites

In the maps for all three states, densities were sufficiently well resolved for us to model a portion of the 3' ss/dsDNA junction. In the closed and intermediate states, the double-stranded region of the DNA is resolved extending from the bottom of the central chamber of RFC through to PCNA, while the 5' overhang on the template strand passes through the opening between the AAA+ and A' domains of Rfc1 (*Figure 3A and B* and *Figure 3—figure supplement 1*). When bound at this position, which we call site 1, the double-stranded region of the DNA in the central chamber is coordinated by Rfc1, Rfc4, Rfc3, and Rfc2. The backbone of the template strand binds to conserved isoleucine and arginine residues on helix α5 of Rfc4, Rfc3, and Rfc2 and to the side chains of Rfc1-Ser384 and Rfc1-Thr386, while the backbone of the primer strand binds to the side chain of Rfc5-Asn80 (*Figure 3—figure supplement 1*). The side chain of Rfc1-Arg434 is inserted into the minor groove and interacts with the bases from both strands. Despite the surface of PCNA containing an abundance of positively charged arginine and lysine residues, we do not observe any direct coordination of the dsDNA by PCNA.

The coordination of the double-stranded region in site 1 by RFC guides the 3' ss/dsDNA junction to its binding site on the collar domain of Rfc1. Binding to RFC:PCNA partially melts the double-stranded region of the DNA near the junction and we resolve density for unpaired bases on the primer strand that extend into an opening between the AAA + domains of Rfc1 and Rfc4 that is continuous with the bulk solvent (*Figure 3B* and *Figure 3—figure supplement 1*). The last paired base of the double-stranded region of the primer strand binds to the side chain of Rfc1-Trp638, which serves a role analogous to the separation pin of helicases (*Baretić et al., 2020*; *Büttner et al., 2007*; *Lee and Yang, 2006*; *Manthei et al., 2006*). The unpaired region of the primer strand is stabilized through both hydrophobic and polar interactions. The first unpaired base of the primer strand binds to Rfc1-Phe582, while the phosphate backbone of the second and third unpaired bases are coordinated by Rfc4-Arg272 and Rfc4-Lys275, respectively. Notably, additional density is resolved extending from the last modelled base of the primer strand that could not be modelled, indicating that the number of unpaired bases on the primer strand varies among the imaged particles (*Figure 3—figure supplement 1*). The 5' overhang of the template strand is guided through the opening between the AAA+ and A' domains of Rfc1 through polar and hydrophobic interactions with Rfc5-Asn103, Rfc1-Asn459, Rfc1-Pro461, Rfc1-Arg464, Rfc1-Phe552, Rfc1-Arg632, Rfc1-Gln636, Rfc1-Phe666, Rfc1-Trp669, and Rfc1-Leu670.

In the open state, the 3' ss/dsDNA junction binds between the collar, AAA+, and A' domains of Rfc1, which we call site 2 (*Figure 3C and D* and *Figure 3—figure supplement 2*). The 3' junction binds to a hydrophobic surface at the interface between the collar and A' domains of Rfc1 formed by Phe666, Trp669, and Leu670, with Phe666 serving as the separation pin for the template strand (*Figure 3D*). The unpaired bases on the 5' overhang of the template strand bind to Asn459, Phe552, and Arg663 of Rfc1. While the precise positions of bases cannot be determined due to disorder, we can follow densities corresponding to a portion of the 5' overhang on the template strand in a 5 Å low-pass filtered map as it wraps around the outside of the collar domain of Rfc1 and re-enters RFC through the opening between the AAA+ domains of Rfc1 and Rfc4 (*Figure 3—figure supplement 2*). There, several bases are again sufficiently ordered for modelling, and we observe that two of the bases occupy the same positions as the unpaired bases of the primer strand in the closed and intermediate states, indicating that it is a high-affinity binding site for ssDNA.

Eight base pairs of the double-stranded region of the DNA in site 2 are resolved in the density map, extending from the junction through the gap between the AAA+ and A' domains of Rfc1 toward the opening in the PCNA ring between PCNA$_A$ and PCNA$_C$. The backbone of the template strand in

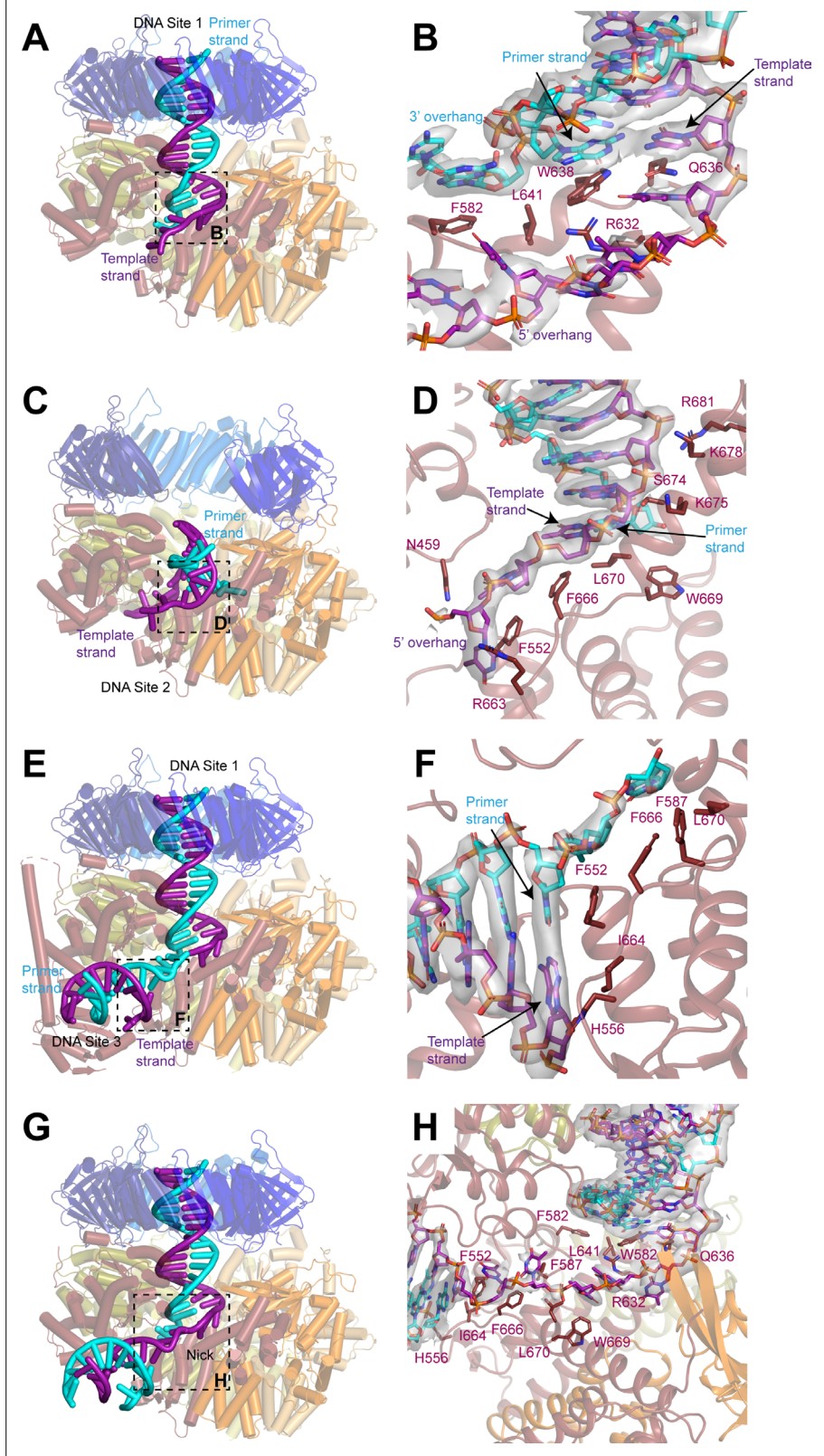

**Figure 3.** DNA binds to three sites in replication factor C (RFC). (**A, C, E**) Structures of RFC:PCNA:DNA in a closed state with a 3' single-stranded (ss)/double-stranded DNA (dsDNA) junction occupying site 1 (**A**), in an open state with a 3' ss/dsDNA junction occupying site 2 (**C**) and in a closed state with 3' ss/dsDNA junctions occupying sites 1 and 3 (**E**). Labels are placed near the 5' ends of the template and primer strands. Regions in dashed boxes

*Figure 3 continued on next page*

*Figure 3 continued*

correspond to panels B, D, and F. (B, D, F) Coordination of 3' ss/dsDNA junctions by RFC at site 1 (**B**), site 2 (**D**), and site 3 (**F**). Arrows point to the last paired bases on the template and primer strands. Density for DNA is shown as gray surface and contoured at 8σ. (**G**) Structure of RFC in a closed state with a nicked dsDNA. Region in dashed box corresponds to panel H. (**H**) Coordination of nicked dsDNA by RFC at sites 1 and 3. Density for DNA is shown as gray surface and contoured at 4σ. PCNA, proliferating cell nuclear antigen.

The online version of this article includes the following source data and figure supplement(s) for figure 3:

**Figure supplement 1.** 3' Single-stranded (ss)/double-stranded DNA (dsDNA) junction binding site 1.

**Figure supplement 2.** 3' Single-stranded (ss)/double-stranded DNA (dsDNA) junction binding site 2.

**Figure supplement 3.** Cryo-EM analysis of RFC:PCNA with DNA substrate 2 (DNA$_2$).

**Figure supplement 4.** Coordination of two 3' single-stranded (ss)/double-stranded DNA (dsDNA) junctions by replication factor C (RFC).

**Figure supplement 5.** Analysis of proliferating cell nuclear antigen (PCNA) loading activity of RFC-1ΔN.

**Figure supplement 5—source data 1.** Uncropped images of gels presented in *Figure 3—figure supplement 5*.

**Figure supplement 6.** Cryo-EM analysis of RFC:PCNA with a nicked DNA.

site 2 is coordinated by Ser674, Lys675, Lys678, and Arg681 on the A' domain of Rfc1 (*Figure 3D* and *Figure 3—figure supplement 2*). While the gap between PCNA$_A$ and PCNA$_C$ is sufficiently wide to accommodate dsDNA, the minimum distance between the AAA+ and A' domains of Rfc1 is only 15 Å. For a DNA to bind at site 2, it must adopt a highly distorted conformation. Indeed, the fourth and fifth bases of the double-stranded region base stack on the opposing strand and do not adopt canonical Watson-Crick base pairing to enable the DNA to access to the narrow opening (*Figure 3—figure supplement 2*).

In several of the maps, in addition to the well-resolved DNA densities at site 1 or 2, weak densities resembling a second segment of dsDNA are present between the AAA+ and collar domains of Rfc1. To better resolve these densities and potentially model a third DNA-binding site, we assembled and collected cryo-EM images of RFC:PCNA in complex with a second DNA substrate that contains a longer, 30 base-pair double-stranded DNA segment and a shorter, 10-base 5' overhang (*Figure 3—figure supplement 3*). Image analysis revealed open, closed, and intermediate classes in the presence of the second DNA substrate at resolutions from 2.8 to 2.1 Å, including open and closed states in which densities for two 3' ss/dsDNA junctions could be resolved (*Figure 3—figure supplement 3* and *Table 1*). One of the DNA molecules occupies site 1 and binds to RFC in a nearly identical fashion as the first substrate does when bound in site 1 (*Figure 3—figure supplement 4*). The densities for the second DNA between AAA+ and collar domains of Rfc1 are much clearer than in the maps with the first DNA substrate and we were able to model to a second 3' ss/dsDNA junction at a position we call site 3 (*Figure 3E and F*, *Figure 3—figure supplement 3* and *Figure 3—figure supplement 4*). When bound to site 3, the double-stranded region of the DNA is partially melted and extends from the junction in a direction perpendicular to the clamp/clamp loader axis (*Figure 3E*). Rfc1 binds to both strands at the junction, with Phe552 interacting with the last base of the primer strand in a manner analogous to separation pins in helicases, while His556 and Ile664 interact with the last base of the template strand (*Figure 3F*). Several bases of the 3' overhang of the primer strand are well resolved and bind to a hydrophobic region of Rfc1 formed by Phe552, Phe587, Phe666, and Leu670 that guides the overhang toward the central chamber of RFC. Extending away from the junction, both strands of the double-stranded region form contacts with residues on the outside of the AAA+ domain of Rfc1. The primer strand is coordinated through interactions with the backbone nitrogen of Arg477 and the side chains of Asn459, Gln474, and Arg477, while the template strand is coordinated by interactions with the backbone of Lys314 and Gly315 (*Figure 3—figure supplement 4*).

## The ss/dsDNA junctions in nicked DNA bind to RFC:PCNA at multiple sites

The 3' end of the primer strand in site 3 is located immediately adjacent to the 5' phosphate of the last resolved base at the 5' end of the template strand in site 1. Together the DNA junctions in sites 1 and 3 thus resemble two segments of dsDNA connected by a short single-stranded segment. RFC has been demonstrated to load PCNA at DNA nicks in vitro, which is relevant for RFC-PCNA function

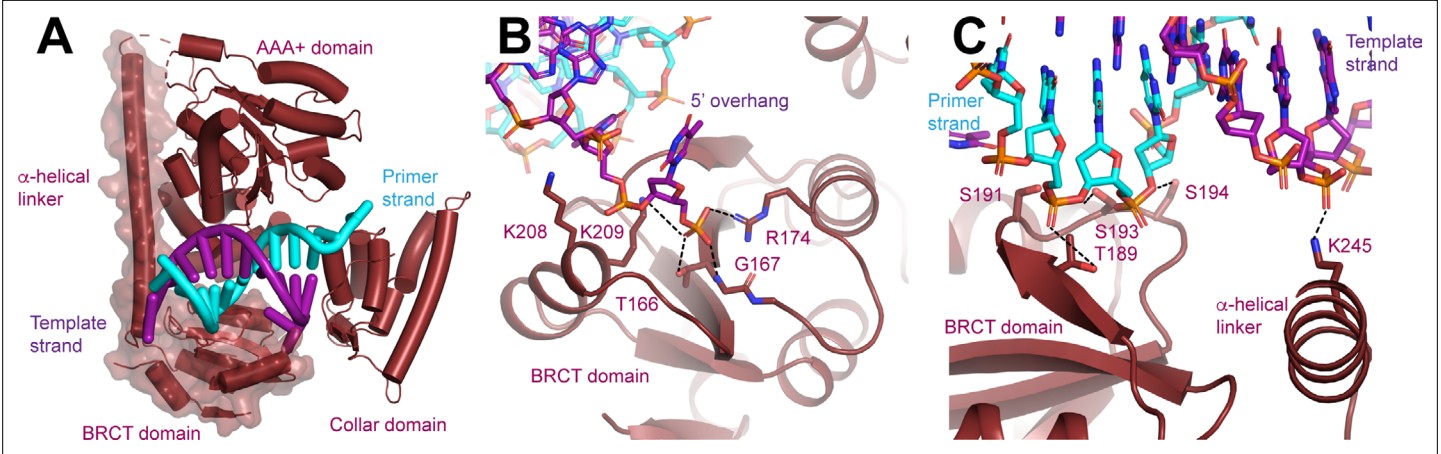

**Figure 4.** N-terminal domain (NTD) of Rfc1. (**A**) Structure of Rfc1 bound to a 3' single-stranded (ss)/double-stranded DNA (dsDNA) junction in site 3. The NTD is shown as a surface and the A' domain is removed for clarity. The dashed line represents the disordered residues between the NTD and AAA+ domain of Rfc1. (**B**) The BRCT domain binds the phosphate of the last ordered base on the 5' overhang of the template strand. (**C**) The BRCT domain and the α-helical linker bind to both strands of a DNA in site 3.

The online version of this article includes the following figure supplement(s) for figure 4:

**Figure supplement 1.** Structure of the Rfc1 N-terminal domain (NTD).

in DNA repair (*Bylund and Burgers, 2005*; *Cai et al., 1996*; *Pluciennik et al., 2006*). We therefore hypothesized that RFC can bind and load PCNA at nicked DNA by binding one of the double-stranded segments to site 1 and the second segment to site 3. We confirmed that our purified RFC could indeed load PCNA on nicked plasmid DNA using a DNA pull-down assay (*Figure 3—figure supplement 5*). To visualize PCNA loading at DNA nicks, we assembled and collected cryo-EM images of RFC:PCNA in complex with a 50 base-pair nicked dsDNA substrate (*Figure 3—figure supplement 6*). Image analysis revealed open and closed states at resolutions from 2.4 to 2.7 Å (*Figure 3—figure supplement 6* and *Table 1*). Among these classes were open and closed states in which the nicked DNA was coordinated with one double-stranded segment in the central chamber at site 1 and the second double-stranded segment between the AAA+ and collar domains at site 3 connected by a single-stranded region (*Figure 3G and H*). These double-stranded segments are coordinated by Rfc1 in an identical fashion to how it coordinates 3' ss/dsDNA junctions in sites 1 and 3 with Trp582 and Gln636 binding to the last paired bases of the double-stranded region of the nicked DNA that occupies site 1 and Phe552, His556, and Ile664 binding the last bases of the double-stranded region of the nicked DNA that occupies site 3 (*Figure 3H*). Moreover, both double-stranded regions of nicked DNA are melted and the 3' and 5' overhangs bind to the same residues of RFC as do the 3' and 5' overhangs of the 3' ss/dsDNA junctions that bind to RFC at sites 1 and 3, respectively. The 5-base single-stranded segment of the nicked DNA connecting the two double-stranded regions is coordinated by interactions with Phe552, Phe587, Arg632, Phe666, Trp669, and Leu670. Thus, RFC employs the same mechanism to load PCNA onto nicked DNA and 3' ss/dsDNA junctions.

## The Rfc1 NTD facilitates PCNA loading at replication forks

Adjacent to the double-stranded portion of DNA in site 3, we observed additional protein densities in a subset of the particles with the second substrate into which we built the NTD of Rfc1, which is absent or poorly resolved in the maps determined with the first and third substrates (*Figure 4A*, *Figure 1—figure supplement 2*, *Figure 3—figure supplement 3*, *Figure 3—figure supplement 6* and *Figure 4—figure supplement 1*). The Rfc1 NTD is composed of a BRCT domain and a long α-helical linker that is connected to the N-terminus of the AAA+ domain by a short, disordered loop (*Figure 4A*). When ordered, the NTD primarily associates with the core of Rfc1 through polar interactions between the α-helical linker and the AAA+ domain of Rfc1. These interactions are quite weak as three-dimensional (3D) classification revealed that even when ordered the NTD can adopt multiple configurations with respect to the AAA+ domain of Rfc1. In contrast, the connection between the

BRCT domain and the α-helical linker seems to be more rigid as they slide along the surface of the AAA+ domain as a single unit (*Figure 4—figure supplement 1*).

The NTD possesses an extended electropositive surface which binds the 5′ overhang of the template strand and the double-stranded region of the DNA as it extends away from RFC (*Figure 4—figure supplement 1*). From the 3′ junction-binding site on the collar domain, the 5′ overhang is guided to the BRCT domain where it binds to Thr166, Gly167, Arg174, Lys208, and Lys209 (*Figure 4B*). Thr166, Gly167, Arg174, and Lys 209, which are all universally conserved among Rfc1 orthologs, form a binding site for the 5′ phosphate of the last modelled base. The phosphate-binding site, which was initially predicted based on comparison of a solution structure of a fragment of the NTD of human Rfc1, is remarkably similar to the structure of the phospho-peptide-binding site of the BRCT domain of BRCA1 (*Clapperton et al., 2006*; *Figure 4—figure supplement 1*).

Extending away from the 3′ ss/dsDNA junction, both strands of the double-stranded section of the DNA in site 3 bind to the Rfc1 NTD (*Figure 3C*). Notably, only the template strand, which binds to the side chain of Arg245, binds to the positively charged side chains of the NTD (*Figure 4C*). The primer strand makes numerous, but weaker contacts with the side chains of Thr189, Ser191, Ser193, and Ser194 and the backbone of Lys190. Together the extensive interactions that the NTD forms with both the 5′ overhang and the double-stranded region of the DNA rationalize the preference of the Rfc1 BRCT domain for binding double-stranded DNA featuring a recessed and phosphorylated 5′ end (*Burbelo et al., 2006*; *Fotedar et al., 1996*; *Kobayashi et al., 1997b*; *Allen, 1997*). The participation of residues in the α-helical linker moreover rationalize why the DNA-binding affinity of a fragment containing residues flanking the human Rfc1 BRCT domain was greatly enhanced compared to the BRCT domain by itself (*Kobayashi et al., 1997b*).

To further evaluate the role of the NTD in loading PCNA onto different DNA substrates, we purified a RFC variant that lacks Rfc1 residues 2–280 (RFC-1ΔN; *Figure 5A*) and compared its activities to that of full-length RFC (RFC-WT). Like RFC-WT, the ATPase activity of RFC-1ΔN is maximally stimulated in the presence of both DNA and PCNA (*Figure 1—figure supplement 1* and *Figure 3—figure supplement 5*), indicating that the catalytic activity of RFC-1ΔN is not grossly affected by the absence of the NTD. To directly observe PCNA loading onto DNA, we monitored the association of PCNA with nicked circular plasmid DNA immobilized via biotin-linkage on paramagnetic streptavidin beads (*Figure 3—figure supplement 5*). In this assay, the association of PCNA with DNA is dependent on both ATP and RFC (*Figure 3—figure supplement 5*), demonstrating that it is due to bona fide PCNA loading onto DNA. Importantly, under these conditions, RFC-1ΔN did not exhibit a noticeable PCNA loading defect, indicating that the Rfc1 NTD is not essential for PCNA loading at DNA nicks, as has also been noted previously (*Pluciennik et al., 2006*). In fact, RFC-1ΔN appeared to be slightly more active for PCNA loading in this assay than RFC-WT. However, as noted before (*Gomes et al., 1997*), full-length RFC has a greater propensity to form aggregates compared to RFC variants lacking the Rfc1 NTD and we, therefore, attribute the reduced PCNA loading activity of full-length RFC in this assay to its potentially enhanced aggregation on the DNA beads. Alternatively, the reduced PCNA loading activity of full-length RFC relative to RFC-1ΔN observed in this assay may be due to the previously noted non-specific dsDNA-binding activity of full-length RFC, which may limit RFC turnover at 3′ ss/dsDNA junctions (*Gomes et al., 1997*; *Uhlmann et al., 1997*; *Podust et al., 1998b*).

To circumvent potential issues due to protein solubility, we compared the PCNA-dependent ATPase activities of RFC-WT and RFC-1ΔN in solution, under dilute conditions (10 nM RFC-WT or RFC-1ΔN) and in the presence of limiting concentrations of a 3′ ss/dsDNA junction substrate. In these conditions, RFC-WT consistently exhibited a greater ATPase activity over the range of DNA concentrations tested than RFC-1ΔN (*Figure 5B*). This data suggests that the Rfc1 NTD indeed promotes PCNA loading at 3′ ss/dsDNA junctions.

We reasoned that a defect in PCNA loading in the absence of the Rfc1 NTD may cause defects in lagging strand synthesis at replication forks as primer extension during Okazaki fragment synthesis is mediated by Pol δ, which is strictly dependent on PCNA for processive DNA synthesis (*Kunkel and Burgers, 2017*). To test this hypothesis, we performed DNA replication reactions on ARS-containing circular plasmid DNA templates in vitro in the absence of RFC or in the presence of varying concentrations of RFC-WT or RFC-1ΔN using the reconstituted budding yeast DNA replication system (*Devbhandari et al., 2017*; *Devbhandari and Remus, 2020*). To differentiate leading and lagging strand products, reactions were carried out in the absence of Cdc9 (DNA ligase 1) and Fen1, which

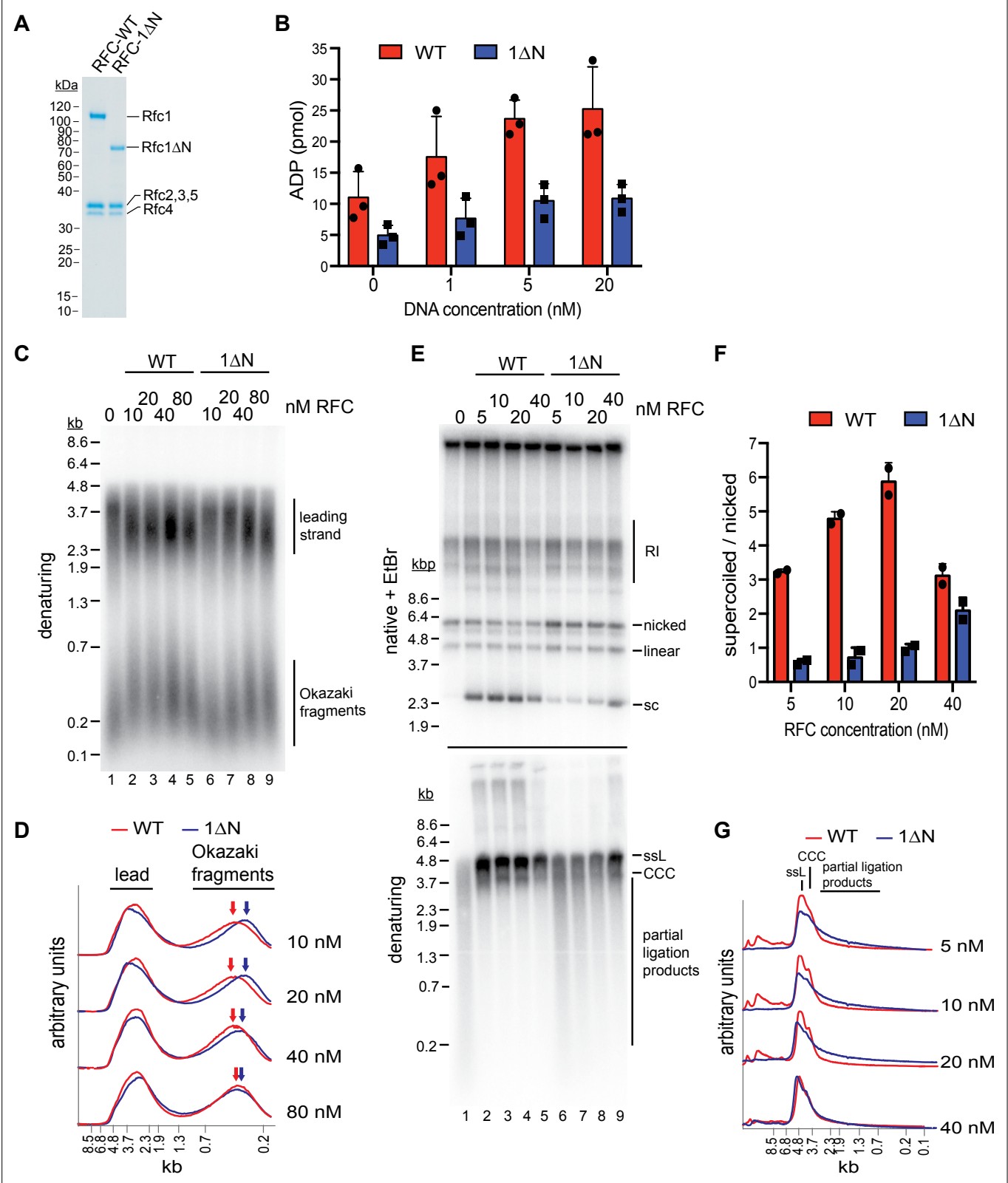

**Figure 5.** The Rfc1 N-terminal domain is required for efficient proliferating cell nuclear antigen (PCNA) loading. (**A**) Representative Coomassie-stained SDS-PAGE analysis of purified RFC-WT and RFC-1ΔN. (**B**) PCNA-dependent ATPase activity of RFC-WT (red bars) or RFC-1ΔN (blue bars) in the presence of varying DNA concentrations. Data are shown in triplicate. Error bars represent standard deviation. (**C**) Denaturing gel analysis of replication products obtained in the absence (lane 1) or presence of variable concentrations of RFC-WT (lanes 2–5) or RFC-1ΔN (lanes 6–9). Reactions were performed in the

*Figure 5 continued on next page*

*Figure 5 continued*

absence of Cdc9 and Fen1. (**D**) Lane traces of gel in panel C, lanes 2–5 (red, RF-WT) and lanes 6–9 (blue, RFC-1ΔN). (**E**) Replication products obtained in the presence of Cdc9 and Fen1 were analyzed by native agarose gel electrophoresis in the presence of ethidium-bromide (EtBr, top) or denaturing gel electrophoresis (bottom). RFC-WT (lanes 2–5) or RFC-1ΔN (lanes 6–9) were included at concentrations indicated on top. RI: replication intermediates; sc: supercoiled; ssL: single-stranded linear; CCC: covalently closed circle. (**F**) Ratio of supercoiled (fully ligated) to nicked (partially ligated) replication products observed by native agarose gel analysis in presence of EtBr as in panel E. Data are shown in duplicate. Error bars represent standard deviation. (**G**) Lane traces of replication products observed in denaturing gel analysis of panel E. Respective concentrations of RFC-WT (red) and RFC-1ΔN (blue) are indicated on the right. All assays were replicated at least two times. RFC, replication factor C.

The online version of this article includes the following source data for figure 5:

**Source data 1.** Uncropped images of gels presented in *Figure 5*.

prevents nascent strand ligation (*Figure 5C and D*). As we have shown previously (*Devbhandari et al., 2017*), in the absence of RFC, Okazaki fragments are synthesized by Pol α or Pol α and Pol ε, resulting in a globally reduced Okazaki fragment length. As expected, addition of RFC-WT at a concentration as low as 10 nM resulted in a marked increase in Okazaki fragment length, consistent with Pol δ carrying out the bulk of primer extension on the lagging strand. In contrast, full Okazaki fragment extension required four to eight times higher concentrations of RFC-1ΔN, indicating impaired Pol δ activity due to a PCNA loading defect in the absence of the Rfc1 NTD. Moreover, we note that leading strands, which normally terminate at the point of fork convergence at the plasmid pole opposite the replication origin, on average reached greater than half-unit lengths in the absence of RFC or at low concentrations of RFC-1ΔN. Since leading strand synthesis terminates at the 5' end of the lagging strand of an opposing fork (*Dewar and Walter, 2017*), this effect could further indicate a defect in lagging strand synthesis in the absence of full RFC activity, allowing leading strands to be synthesized significantly beyond the point of sister replisome convergence. Alternatively, since PCNA is known to stabilize Pol ε on the leading strand (*Kumar et al., 2021*; *Yeeles et al., 2017*), the stochastic stalling or slowing of one of the two forks emanating from the replication origin may shift the site of termination beyond the midpoint between the two opposing forks. Irrespective, the data is consistent with the Rfc1 NTD promoting PCNA loading at replication forks.

Since the strand-displacement activity of Pol δ, which is absent in Pol α and Pol ε, is essential for Okazaki fragment ligation (*Devbhandari et al., 2017*), defects in PCNA-dependent Pol δ activity are expected to result in a corresponding defect in Okazaki fragment ligation. To test this prediction, we carried out in vitro replication reactions in the presence of Cdc9 and Fen1 (*Figure 5E–G*). In the absence of RFC, nascent strand ligation is severely compromised, as evidenced by the absence of covalently closed plasmid daughters, which can be detected as plasmid supercoils by native agarose gel analysis in the presence of ethidium-bromide (EtBr), and the presence of largely unligated nascent strands in the denaturing gel analysis. In contrast, as little as 5 nM RFC-WT results in the formation of a significant fraction of covalently closed plasmid daughters and the generation of full-length or near full-length nascent strands in the denaturing gel analysis, indicative of efficient nascent strand ligation. Conversely, native gel analysis of the replication products obtained after addition of 5–40 nM RFC-1ΔN results in a greatly reduced formation of covalently closed plasmid daughters and a concomitant increase in nicked or gapped plasmid daughters, while a prominent smear of partially ligated nascent strands is evident across the same range of RFC-1ΔN concentrations by denaturing gel analysis. In summary, we conclude that the Rfc1 NTD mediates efficient PCNA loading at replication forks, which is required for normal Okazaki fragment synthesis and ligation.

## Discussion

In previous structural studies, the RFC:PCNA complex was captured in an autoinhibited state off DNA (*Bowman et al., 2004*; *Gaubitz et al., 2020*). Moreover, the flexible NTD of Rfc1 was deleted in those studies to improve protein yield and solubility, precluding its structural and functional characterization. Therefore, to advance our insight into the PCNA loading process, we have reconstituted RFC:PCNA:DNA complexes bound to ATPγS with full-length proteins and analyzed their structure using cryo-EM, which allowed us to capture RFC:PCNA in an active state. We observe that PCNA can adopt both open and closed conformations in the presence of RFC, DNA, and ATPγS. The open state is consistent with previous biochemical studies demonstrating that ATP binding, but not ATP hydrolysis,

is required for PCNA opening (*Zhuang et al., 2006*; *Johnson et al., 2006*; *Yao et al., 2006*; *Gomes and Burgers, 2001a*; *Hingorani and Coman, 2002*). Unexpectedly, however, the PCNA ring opens in a largely planar fashion, generating a gap of ~22 Å that aligns with the gap in RFC between the AAA+ and A' domains of Rfc1 and is wide enough for the passage of dsDNA. The wide opening of PCNA during loading is similar to that which we observed for the 9-1-1 checkpoint clamp by Rad24-RFC and also consistent with MD simulations and biochemical FRET studies suggesting that PCNA opens an ~30 Å gap during loading (*Zhuang et al., 2006*; *Castaneda et al., 2022*; *Adelman et al., 2010*; *Tainer et al., 2010*). However, the wide opening of PCNA refutes a previous model implicating the adoption of a right-handed spiral conformation with a <10 Å gap by PCNA that was inspired by the spiral configuration of the RFC AAA+ domains observed in the crystal structure of autoinhibited RFC bound to PCNA and supported by MD simulations on the isolated PCNA clamp (*Bowman et al., 2004*; *Kazmirski et al., 2005*). A spiral configuration for PCNA may be more prevalent in the absence of DNA, as our data suggests that the planar RFC-PCNA conformation is stabilized by DNA, which we discuss below. In any case, the open PCNA state observed here is compatible with the direct loading of PCNA around dsDNA and obviates the need to invoke ssDNA threading for RFC:PCNA positioning at 3' ss/dsDNA junctions, which agrees with the ability of RFC to load PCNA at DNA structures lacking extensive stretches of ssDNA, such as DNA nicks.

Closure of the PCNA ring occurs through a two-step process involving release of one protomer from Rfc5 and the concerted inward movement of a second protomer together with Rfc1. Notably, ATPγS remains actively coordinated in all four RFC catalytic sites in the closed PCNA state, demonstrating that PCNA closure is not mechanistically coupled to ATP hydrolysis. Instead, the open and closed states of PCNA appear to exist in a dynamic equilibrium that is driven by the binding energies of RFC-PCNA contacts. As kinetic studies have suggested that ATP hydrolysis precedes PCNA closure (*Chen et al., 2009*; *Sakato et al., 2012b*), it is, therefore, possible that ATP hydrolysis will drive PCNA ring closure by disrupting the wide-planar conformation of RFC. In addition, analogously to Rad24-RFC, we anticipate that ATP hydrolysis will induce PCNA release via an out-of-plane motion of Rfc1 (*Castaneda et al., 2022*). However, due to potential differences in the conformational landscape of RFC-PCNA in the presence of non-hydrolyzable ATPγS and that of RFC-PCNA in the presence of hydrolyzable ATP (*Zhuang et al., 2006*; *Chiraniya et al., 2013*), future studies will be necessary to understand the precise role of ATP hydrolysis by RFC and how hydrolysis is coupled to conformational change and PCNA release.

Our observation that Rfc5 in both RFC and Rad24-RFC binds GDP instead of ADP was highly unexpected, as AAA+ proteins generally utilize ATP and ATP is sufficient to fuel clamp loading by both RFC and Rad24-RFC. Moreover, as no guanine nucleotides were added during either purification, the GDP is likely to have been co-purified with Rfc5 and thus be a very high affinity ligand. To date, only one other AAA+ protein, the McrB motor subunit of the *E. coli* McrBC restriction endonuclease complex, is known to utilize GTP (*Niu et al., 2020*). In the case of Rfc5, future studies will be necessary to determine if Rfc5 is a functional GTPase or if GDP instead serves a structural role.

The Rfc2-5 core can associate with four distinct large subunits, respectively, to form RFC (RFC), which contains Rfc1, or three RFC-like complexes, Ctf18-RFC, Rad24-RFC, and Elg1-RFC (*Lee and Park, 2020*). Each clamp loader complex performs specific functions in the cell that are associated with a distinct DNA substrate specificity. For example, RFC loads PCNA specifically at 3' ss/dsDNA junctions and DNA nicks (*Bylund and Burgers, 2005*; *Cai et al., 1996*; *Ellison and Stillman, 2003*; *Hayner et al., 2014*), Ctf18-RFC loads PCNA at 3' ss/dsDNA junctions but not at DNA nicks (*Bylund and Burgers, 2005*; *Bermudez et al., 2003*), Rad24-RFC loads the 9-1-1 clamp at 5' ss/dsDNA junctions (*Ellison and Stillman, 2003*; *Castaneda et al., 2022*; *Majka et al., 2006*), and Elg1-RFC appears to unload PCNA from dsDNA (*Kubota et al., 2015*). It is, therefore, likely that the DNA substrate specificity of clamp loaders is determined by DNA-protein contacts involving the large subunit. Consistent with this notion, we have recently found that the 5' ss/dsDNA junction is exclusively coordinated by Rad24 in Rad24-RFC (*Castaneda et al., 2022*). Similarly, we find here that 3' ss/dsDNA junctions are exclusively coordinated by Rfc1 in RFC. However, unlike in Rad24-RFC:9-1-1, where the DNA binds to a single site, we observe that DNA binds to three distinct sites in RFC:PCNA. In each case, the 3' ss/dsDNA junctions are positioned at distinct sites along a hydrophobic ridge on the surface of the Rfc1 collar domain. Intriguingly, several base pairs of the 3' ss/dsDNA junction are melted at each of these sites, utilizing distinct separation pin residues. While DNA junction unwinding by RFC was recently

reported at site 1 (*Gaubitz et al., 2022*), it was not observed in the Rad24-RFC:9-1-1:DNA complex (*Castaneda et al., 2022*) and thus may be unique feature of RFC. Because melting is independent of ATP hydrolysis, it does not involve a helicase mechanism. Instead, it appears to be promoted by hydrophobic-binding forces between DNA bases and hydrophobic side chains on the surface of RFC. As described below, our data suggests that junction unwinding drives the insertion of the DNA duplex into the central RFC channel at an intermediate step. In this regard, junction unwinding may aid RFC:PCNA binding particularly at DNA nicks or short gaps by extending the ssDNA stretch in the template strand. Notably, Phe582 and Trp638, which promote base flipping at site 1, are not essential for growth in yeast and thus future studies will be required to understand the precise role of DNA base flipping and unwinding by RFC (*Gaubitz et al., 2022*).

DNA site 1 is located in the center of the clamp loader, analogous to the binding sites resolved in the structures of the bacteriophage T4 and *E. coli* clamp loaders (*Kelch et al., 2011*; *Simonetta et al., 2009*). In this site, the phosphate backbone of the DNA duplex template strand forms interactions with conserved residues on all five clamp loader subunits, including those in the Rfc2-5 core that coordinate the ssDNA in Rad24-RFC, explaining how the Rfc2-5 core can be adapted to bind either ssDNA or dsDNA (*Castaneda et al., 2022*). Moreover, we find that RFC can bind dsDNA in its central channel because Rfc1 lacks the loop that protrudes from the AAA+ domain of Rad24 that occludes dsDNA from binding in the central channel of Rad24-RFC. In agreement with the notion that the PCNA clamp exists in a dynamic equilibrium between open and closed states prior to ATP hydrolysis, we observe both states when DNA is present in the central channel. DNA binding at site 1 thus represents the final stage of the DNA loading reaction.

The second site is located between the AAA+, collar, and A' domains of Rfc1. When bound to this site, the double-stranded portion of the DNA extends between the AAA+ and A' domains of Rfc1 toward the opening in PCNA and thus can only be occupied in the open state. DNA site 2, therefore, corresponds to the DNA-binding site at an intermediate state that precedes occupation of DNA site 1. Interestingly, the gap between the Rfc1 AAA+ and collar domains is not wide enough to accommodate normal B-form DNA, causing a significant distortion in the DNA duplex as it transits into the central RFC channel. The energetic penalty associated with the dsDNA distortion indicates that the gap between the AAA+, collar, and A' domains of Rfc1 acts as a gate. Rotation of the DNA induced by the unwinding of the 3' ss/dsDNA junction at the base of the gate may help push the DNA through the constriction of the gate, aided by the conformational flexibility of the open interface between the AAA+, collar, and A' domains.

The third binding site is similar to the DNA junction-binding site resolved in Rad24-RFC:9-1-1:DNA (*Castaneda et al., 2022*), located between the AAA+ and collar domains of Rfc1 and with the double-stranded region of the DNA extending away from the clamp loader. Also contributing to the third binding site is the BRCT domain of Rfc1, which contacts the phosphate backbone of the double-stranded DNA. Previous studies have demonstrated that the BRCT domain of RFC1 forms an autonomous dsDNA-binding domain (*Burbelo et al., 2006*; *Fotedar et al., 1996*; *Kobayashi et al., 1997b*; *Allen, 1997*). The BRCT domain is anchored on the side of the Rfc1 AAA+ domain via a long linker helix and bridging DNA contacts. The BRCT domain and linker helix appear to form a rigid unit that is flexibly tethered to the AAA+ domain via a short linker. The flexible association of the BRCT domain promotes its ability to explore the 3D space around RFC for DNA targets, thus promoting the recruitment of RFC to the DNA substrate. Accordingly, while not being essential for RFC:PCNA binding to DNA at non-limiting concentrations of RFC or DNA substrate, the Rfc1 NTD is expected to increase the efficiency of RFC:PCNA:DNA complex formation. Consistent with this notion, we observe that the Rfc1 NTD promotes Okazaki fragment synthesis at replication forks, which likely requires rapid turnover of RFC. In addition, we note that simultaneous coordination of the DNA substrate at multiple DNA-binding sites distributed across the BRCT, AAA+, and collar domains of Rfc1 may help restrain the conformational flexibility of Rfc1 and thus promote the active and planar conformation of RFC-PCNA, which in turn may underlie the observation that both DNA and PCNA stimulate ATP binding to RFC (*Chen et al., 2009*; *Gomes et al., 2001b*; *Sakato et al., 2012b*). DNA site 3 thus appears to be involved in the early stages of RFC:PCNA:DNA complex formation.

Among several weak BRCT-DNA contacts, which may aid a role for the BRCT domain in DNA scanning and handover to DNA site 2, we also observe discrete hydrogen bonds between a highly conserved TG motif of the BRCT domain and a backbone phosphate in the template strand near the

3' ss/dsDNA junction. This interaction may be responsible for the increased affinity of the Rfc1 BRCT domain for 5'-phosphorylated dsDNA fragments noted previously (*Kobayashi et al., 1997b*; *Allen, 1997*) and may help position and stabilize RFC at the junction. Notably, the interaction of the TG motif with the DNA phosphate backbone appears to be conserved in the homologous PARP1 BRCT domain (*Rudolph et al., 2021*). As has been noted previously, residues corresponding to the Rfc1 TG motif in tandem BRCT domains mediate the interaction with the phosphate moiety of phospho-peptides, indicating that the phospho-peptide-binding activity of tandem BRCT domains has evolved from the ancestral DNA-binding activity of single BRCT domains (*Kobayashi et al., 1997a*; *Kobayashi et al., 1997b*).

Our data suggest the following model for PCNA loading (*Figure 6*): RFC-PCNA initially adopts an autoinhibited and partially engaged conformation that searches for a DNA substrate using the flexible NTD of Rfc1 to probe the surrounding space. Binding of DNA to the BRCT domain then promotes occupation of DNA site 3 and induction of an active planar RFC:PCNA conformation. While PCNA can adopt an open configuration when bound to RFC in the absence of DNA (*Gaubitz et al., 2022*), adopting the active planar conformation shifts the equilibrium between the closed to open states of PCNA toward the open state. Driven by binding energy, the DNA substrate is released from DNA site 3 and subsequently transfers to DNA site 2 to initiate insertion into the central RFC:PCNA channel. By sliding and rotating over the hydrophobic surface of the Rfc1 collar domain, the DNA shifts into position at DNA site 1. DNA binding to site 1 then promotes ATP hydrolysis by RFC, which promotes PCNA closure by disrupting the planar RFC:PCNA interaction surface and consequently the dynamic interaction of PCNA with Rfc2 and -5 in the open state. In addition, ATP hydrolysis is expected to eject PCNA from RFC via an out-of-plane motion of Rfc1. Finally, inactivated RFC releases from the DNA and re-enters the cycle by recruiting PCNA free in solution.

In our model, a step-wise RFC-DNA-binding mechanism mediates the loading of PCNA at 3' ss/dsDNA junctions. While not being essential for RFC-DNA binding, the Rfc1 NTD promotes the initial interaction of RFC with DNA. Based on the MMS sensitivity of yeast cells lacking the Rfc1 NTD (*Gomes et al., 1997*), which may indicate a defect in the post-replicative processing of nicks or gaps in the DNA, we had initially hypothesized that the NTD is specifically required for PCNA loading at DNA nicks. However, we are unable to detect a defect in PCNA loading at DNA nicks in vitro in the absence of the Rfc1 NTD, which is consistent with previous studies demonstrating the proficiency of RFC lacking the Rfc1 NTD in loading PCNA at DNA nicks during mismatch repair in vitro (*Pluciennik et al., 2006*; *Dzantiev et al., 2004*). We, therefore, hypothesize that PCNA loading at DNA nicks follows the same step-wise DNA-binding mechanism as at 3' ss/dsDNA junctions with a single-stranded 5' overhang. However, our data do not discriminate between a transition of the 3' double-stranded segment from site 3 to site 2 or its direct binding of the second double-stranded region in site 2, while maintaining the interaction of the 5' segment in site 3. Through either pathway, RFC-PCNA will be bound to both the 3' and 5' junction of a DNA nick, as observed in our structural analysis here. Future studies will be necessary to elucidate the precise conformational transitions that nicked and gapped DNA undergo to yield the doubly bound, pre-ATP hydrolysis states that we captured.

In conclusion, our study reveals the structural basis for the loading of PCNA at 3' ss/dsDNA junctions by RFC, providing a framework for future studies aimed at understanding the mechanism of PCNA loading and unloading by RFC-like complexes.

## Materials and methods
### Protein expression and purification
PCNA, RFC, and RPA were purified as previously described (*Devbhandari et al., 2017*). RFC-1ΔN lacks Rfc1 residues 2–280 and was purified identically to wildtype RFC using strain *S. cerevisiae* YJC17 (*MATa ade2-1 ura3-1 his3-11,15 trp1-1 leu2-3,112 can1-100 pep4::kanMX bar::hphNAT1 his3::-Gal-Gal4 trp1::Gal-Rfc2-Rfc3 ura3::Gal-Rfc4-Rfc5 leu2::Gal-Rfc1-ΔN*).

### Preparation of 3' ss/dsDNA template
Oligos were purchased from IDT. To generate a DNA substrate with a 20 base-pair duplex and 50 nucleotide overhang, oligo DR2630 (5'-TTTTTTTTTTTTTTTTTTTTTTTTTTTTTTTTTTTTTTTTTTTTTT TTTTTGCGAGGAAGGACTGAGCAGG-3') was annealed to oligo DR2631 (5'-CCTGCTCAGTCCT

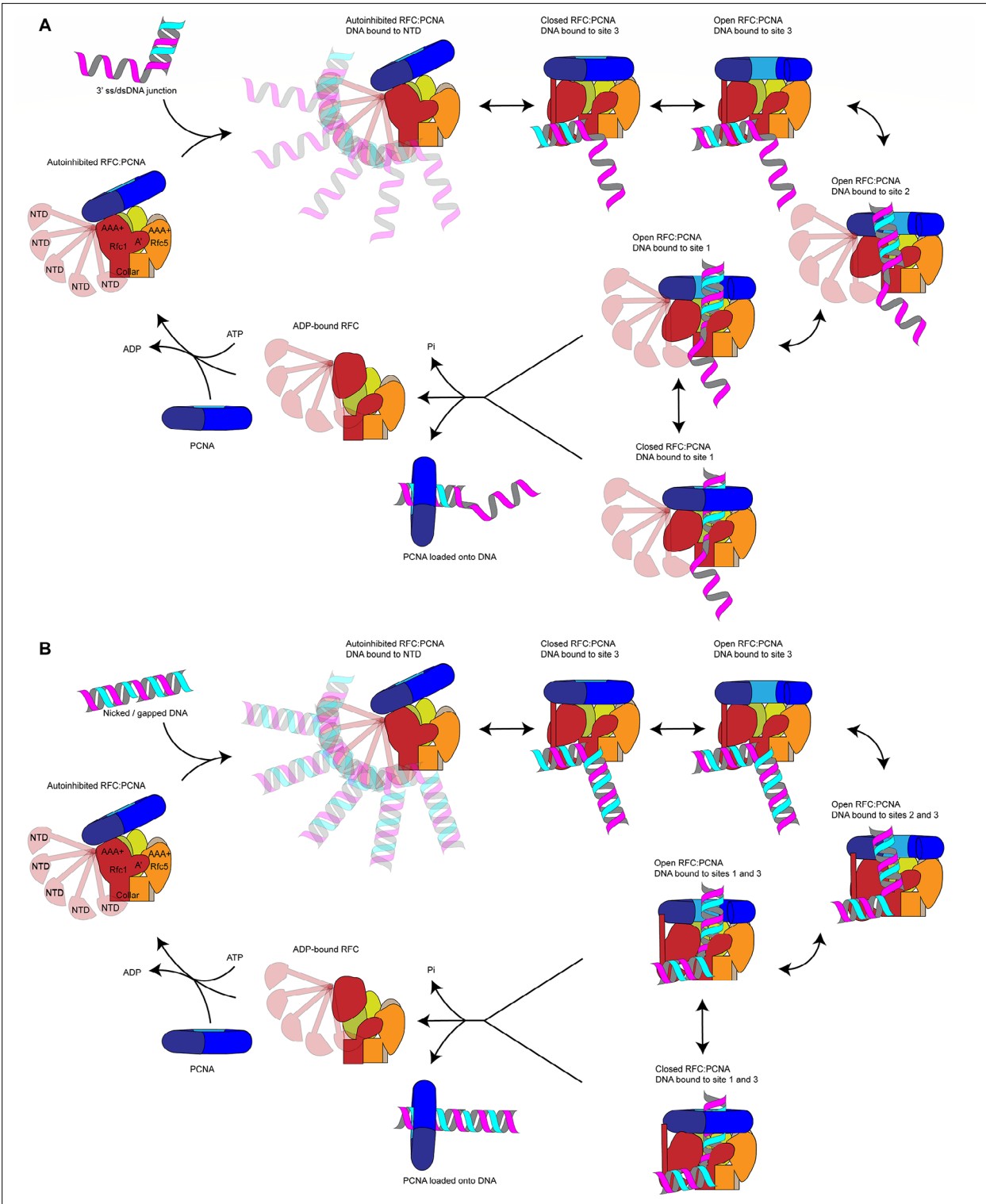

**Figure 6.** Loading for proliferating cell nuclear antigen (PCNA) onto DNA by replication factor C (RFC). (**A**) Model for PCNA loading onto 3′ single-stranded (ss)/double-stranded DNA (dsDNA) junctions. (**B**) Model for PCNA loading onto nicked or gapped DNA. RFC and PCNA are colored by subunit as in *Figure 1*.

TCCTCGC-3'). To generate a DNA substrate with a 30 base-pair duplex and 10 nucleotide overhang, oligo DR2667 (5'-TTTTTTTTTTGCGAGGAAGGACTGAGCAGGCTTCCATACC-3') was annealed to oligo DR2668 (5'-GGTATGGAAGCCTGCTCAGTCCTTCCTCGC-3'). To create a nicked DNA substrate, oligo DR2764 (5'- TAGCCTGCTCAATCCGTCCTCGCCGCTCCGTCCTGACTCGTCCGA-3') was annealed to oligos DR2765 (5'- CGCTCCGTCCTGACTCGTCCGA-3') and DR2766 (5'- CGCTCCGTCCTGACTCGTCCGA-3'). DR2630 and DR2764 were PAGE purified prior to template generation. The DNA oligos were annealed by mixing to a final concentration of 10 µM in 10 mM Tris, pH 7.5/50 mM NaCl/1 mM EDTA, heat denaturation at 95°C, and gradually decreasing the temperature from 95°C to 10°C in steps of 1°C per minute using a thermocycler.

## ATPase assays

Reactions were carried out in buffer containing 5 mM HEPES-KOH, pH 7.6/100 mM KOAc/2 mM Mg(OAc)$_2$/5% glycerol/0.02% NP40S/1 mM DTT. RFC or RFC-1ΔN, PCNA, and 3' ss/dsDNA substrate were included at 10, 100, and 10 nM, respectively, as indicated. Reactions were incubated at 30°C and at the indicated times 2 µL aliquots were spotted on TLC PEI Cellulose F (Millipore) to stop the reaction. The TLC plates were developed in 0.6 M sodium phosphate buffer, scanned on a Typhoon FLA 7000 phosphoimager, and ATP hydrolysis quantified using ImageJ. For DNA titration experiments, 3' ss/dsDNA template was included at 20, 5, and 2.5 nM, as indicated; the reaction time was 5 min.

## Preparation of bead-bound DNA template

Sixteen µg of plasmid DNA (p1285, 8.6 kbp) were nicked with Nb.BbvCI for 2 hr. Nicked DNA was phenol-chloroform extracted and subsequently biotinylated via UV crosslinking following the manufacturer's instructions (Vector Laboratories, SP-1020) with the following modifications: DNA was crosslinked at 365 nm for 20 s. Two mg Dynabeads M-280 Streptavidin (11205D) were washed once in 400 µL buffer containing 25 mM HEPES-KOH pH 7.6/1 mM EDTA/2 M NaCl and resuspended in 400 µL of the same buffer. The biotinylated DNA was added to the beads and incubated overnight at room temperature with rotation. Beads were washed twice with 400 µL buffer containing 25 mM HEPES-KOH pH 7.6/1 mM EDTA/1 M KOAc and once with 400 µL buffer containing 25 mM HEPES-KOH pH 7.6/1 mM EDTA. Beads were resuspended in 200 µL of 25 mM HEPES-KOH pH 7.6/1 mM EDTA and stored at 4°C.

## PCNA loading assay

Reactions were carried out in a 15 µL volume in buffer containing 25 mM HEPES-KOH pH 7.6/100 mM KOAc/5 mM Mg(OAc)$_2$/5% glycerol/0.01% NP40S/1 mM DTT. Ten µL of DNA beads were used per reaction. RFC-WT or RFC-1ΔN were used at concentrations as indicated, while PCNA was included at 120 nM. 1 µL of the reaction was taken as an input fraction. Reactions were incubated at 30°C for 5 min and terminated by addition of 3 µL 0.5 M EDTA, pH 8.0. Unbound fraction was separated from beads using a magnetic rack and discarded, beads were washed twice with 500 µL buffer containing 25 mM HEPES-KOH pH 7.6/300 mM KOAc/5 mM Mg(OAc)$_2$/5% glycerol/0.02% NP40S. Finally, beads were resuspended in 30 µL 1× Laemmli buffer, boiled, and analyzed by SDS-PAGE. For Western blot analysis, membranes were probed with monoclonal anti-FLAG-HRP (Sigma) to detect Rfc1 or polyclonal anti-PCNA (871; *Zhang et al., 2000*).

## Preparation of samples for cryo-EM grids

All reactions were carried out at 30°C. First, PCNA (4.28 µM) and RFC (2.5 µM) were mixed for 30 min in a 49 µL reaction containing 25 mM HEPES-KOH pH 7.6/300 mM KOAc/10% glycerol/1 mM DTT/2.04 mM ATPS/5.1 mM Mg(OAc)$_2$. Subsequently, RPA (30 nM) and 3' ss/dsDNA (1.5 µM) were added to the reaction, the reaction volume was increased to a final volume of 100 µL with reaction buffer (25 mM HEPES-KOH pH 7.6/300 mM KOAc/7 mM Mg(OAc)$_2$/5% glycerol/0.02% NP40S/1 mM DTT) and incubation continued for 30 min. The reaction was layered onto a 4 mL 10–35% glycerol gradient containing 25 mM HEPES-KOH pH 7.6/300 mM KOAc/7 mM Mg(OAc)$_2$/5% glycerol/0.02% NP40S/1 mM DTT. Gradients were centrifuged at 45,000 rpm for 6 hr at 4°C in a Thermo Scientific AH-650 swing bucket rotor. Two-hundred µL fractions were manually collected from the top of the gradient and 10 µL of each fraction analyzed by SDS-PAGE stained with SilverQuest Staining Kit (Invitrogen). Another 10 µL of each fraction were subjected to DNA quantification using a QuBit

3.0 Fluorometer with QuBit dsDNA HS Assay kit (Q32851). Peak fractions were pooled and dialyzed against buffer containing 25 mM HEPES-KOH, pH 7.6/300 mM KOAc/7 mM Mg (OAc)$_2$. Dialyzed sample was concentrated via centrifugation in Amicon Ultracel 30 K 0.5 mL filter units (UFC503096) and the protein concentration of the final sample determined by SDS-PAGE and Coomassie stain by comparing with a known protein standard.

## Replication assays

4 nM ARS1-containing circular plasmid template (p1017, 4.8 kbp) was incubated with 20 nM ORC, 20 nM Cdc6, and 60 nM Cdt1·Mcm2-7, in the presence of 100 mM KOAc, 25 mM HEPES-KOH pH 7.5, 5% glycerol, 2.5 mM DTT, 0.02 % NP-40, and 5 nM ATP for 20 min at 30°C; 150 nM of DDK was added and incubation continued for 20 min at 30°C. Subsequently, a mastermix of proteins was added to yield final concentrations of 0.2 μg/μL BSA, 20 nM Sld3·7, 125 nM Cdc45, 80 nM each dNTP, 16 nM CDK (Clb5-Cdc28), 100 nM GINS, 30 nM Pol ε, 30 nM Dpb11, 30 nM Sld2, 120 nM RPA, 60 nM Pol α, 35 nM Ctf4, 75 nM PCNA, 4 nM Pol δ, 25 nM Csm3-Tof1, 25 nM Mrc1, 0.2 mM rNTP, 30 nM Top1, 30 nM Top2, 0.04 mg/mL creatine kinase, 16 mM creatine phosphate, and 5 μCi α-[$^{32}$P]-dATP was added to the reaction. The salt concentration was increased to 190 mM KOAc using a 3 M KOAc stock solution, RFC or RFC-1ΔN was added at concentrations as indicated in the figure, and reactions were initiated by the addition of 14 nM Mcm10. Fen1 and Cdc9 were included at 15 nM each, as indicated. Reactions were incubated at 30°C for 30 min and stopped by addition of 40 mM EDTA, 1.6 U Proteinase K, and 0.25% SDS, followed by incubation for 30 min at 37°C. DNA was isolated from the reactions by phenol/chloroform extraction and filtration through Illustra MicroSpin G-25 spin columns (GE).

For denaturing gels, 0.8% agarose gels were cast in buffer containing 30 mM NaOH and 2 mM EDTA and 10 μL aliquots of the reactions were fractionated at 45 V for 3 hr in 30 mM NaOH. Gels were neutralized and fixed in 5% TCA. For native gels, 10 μL aliquots of the reactions were fractionated on 0.8% agarose gels containing 0.6 mg/mL EtBr, at 50 V for 3 hr in TAE. Gels were imaged using Typhoon FLA 7000.

Quantification and lane traces of gel images were performed using ImageJ. For *Figure 5F*, intensities of nicked and supercoiled products were computed relative to total intensity in each lane. The ratio of supercoiled to nicked products was then tabulated in GraphPad Prism.

## Cryo-EM sample preparation and data acquisition

For RFC:PCNA in the complex with each of the three DNA substrates, 3.5 μL of purified protein at a concentration of 0.22 mg/mL was applied to Graphene Oxide Au 400 mesh QUANTIFOIL R1.2/1.3 holey carbon grids (Quantifoil), and then plunged into liquid nitrogen-cooled liquid ethane with an FEI Vitrobot Mark IV (FEI Thermo Fisher). The sample was frozen at 4°C with 100% humidity, using blotting times between 30 and 60 s and a waiting time of 30 s. Grids were transferred to a 300 keV FEI Titan Krios microscopy equipped with a K3 summit direct electron detector (Gatan). Images were recorded with SerialEM (*Mastronarde, 2005*) in super-resolution mode at 29,000×, corresponding to super-resolution pixel size of 0.413 Å. Dose rate was 15 electrons/pixel/s, and defocus range was −0.5 to −2.0 μm. Images were recorded for 3 s with 0.05 s subframes (total 60 subframes), corresponding to a total dose of 66 electrons/Å (*Gulbis et al., 1996*).

## Cryo-EM processing

Sixty-frame super-resolution movies (0.413 Å/pixel) of RFC:PCNA in the complex with dsDNA were gain corrected, Fourier cropped by two (0.826 Å/pixel) and aligned using whole-frame and local motion correction algorithms by cryoSPARC v3.2.0 (*Punjani et al., 2017*). Blob-based autopicking in cryoSPARC was implemented to select initial particle images. Several rounds of 2D classification were performed and the best 2D classes were manually selected for the initial 3D model generation using the ab initio algorithm in cryoSPARC. False-positive selections and contaminants were excluded through iterative rounds of heterogeneous classification using the model generated from the ab initio algorithm, followed by particle polishing in Relion 3.1.2 (*Scheres, 2016*). The polished particles were then classified using 3D classification in cryoSPARC v3.3.1 (*Punjani et al., 2020*). Once classification had converged, the final particle stacks were refined using non-uniform refinement in cryoSPARC v3.3.1 with local CTF estimation and higher order aberration correction. The reconstructions were

further improved by employing density modification on the two unfiltered half-maps with a soft mask in Phenix (*Terwilliger et al., 2020*).

## Model building and refinement

The structures of the yeast RFC-PCNA complex (PDB: 1SXJ) (*Bowman et al., 2004*) were manually docked into the closed structure of RFC:PCNA with DNA substrate 1 in UCSF Chimera (*Pettersen et al., 2004*). The models were manually rebuilt to fit the density and sequence in COOT (*Emsley et al., 2010*). The models were initially refined in ISOLDE (*Croll, 2018*) to correct geometric errors before several cycles of manual rebuilding in COOT and real space refinement in Phenix (*Liebschner et al., 2019*) against the closed state map.

The refined closed state model was manually docked into the open and intermediate state maps determined with DNA substrate 1 and manually rebuilt to fit the density in COOT. The final models were subjected to real space refinement in Phenix against the open state map. For DNA substrates 2 and 3, the appropriate DNA substrate 1 model was docked into the density maps, rebuilt in COOT, and refined using real space refinement in Phenix.

Figures were prepared using PyMol (http://www.pymol.org/), APBS (*Jurrus et al., 2018*), UCSF Chimera *Pettersen et al., 2004*, and UCSF ChimeraX *Goddard et al., 2018*.

## Materials availability statement

All plasmids and strains are available from the corresponding authors after completing a material transfer agreement.All plasmids and strains are available from the corresponding authors after completing a materials transfer agreement.

## Acknowledgements

We thank M de la Cruz at the MSKCC Richard Rifkind Center for cryo-EM for assistance with data collection and the MSKCC HPC group for assistance with data processing. This work was supported by NIH-NCI Cancer Center Support Grant P30 CA008748 (DR, RKH), NIGMS R01-GM107239, and NIGMS R01-GM127428 (DR), and the Josie Robertson Investigators Program (RKH). MS is a Walter Benjamin Fellow of the Deutsche Forschungsgemeinschaft. JCC is supported by an NIH T32 training grant (T32GM132081).

## Additional information

### Competing interests

### Funding

| Funder | Grant reference number | Author |
|---|---|---|
| National Cancer Institute | P30 CA008748 | Richard K Hite<br>Dirk Remus |
| National Institute of General Medical Sciences | R01-GM107239 | Dirk Remus |
| National Institute of General Medical Sciences | R01-GM127428 | Dirk Remus |
| National Institute of General Medical Sciences | T32GM132081 | Juan C Castaneda |
| Deutsche Forschungsgemeinschaft | | Marina Schrecker |
| Josie Robertson Investigators Program | | Richard K Hite |

The funders had no role in study design, data collection and interpretation, or the decision to submit the work for publication.

## Author contributions
Marina Schrecker, Conceptualization, Data curation, Formal analysis, Funding acquisition, Validation, Investigation, Visualization, Methodology, Writing - original draft, Writing - review and editing; Juan C Castaneda, Conceptualization, Resources, Data curation, Formal analysis, Validation, Investigation, Visualization, Methodology, Writing - original draft, Writing - review and editing; Sujan Devbhandari, Charanya Kumar, Resources, Validation, Investigation, Visualization, Methodology, Writing - review and editing; Dirk Remus, Richard K Hite, Conceptualization, Resources, Data curation, Formal analysis, Supervision, Funding acquisition, Validation, Investigation, Visualization, Methodology, Writing - original draft, Project administration, Writing - review and editing

## Author ORCIDs
Marina Schrecker http://orcid.org/0000-0001-8542-6657
Dirk Remus http://orcid.org/0000-0002-5155-181X
Richard K Hite http://orcid.org/0000-0003-0496-0669

## Decision letter and Author response
Decision letter https://doi.org/10.7554/eLife.78253.sa1
Author response https://doi.org/10.7554/eLife.78253.sa2

---

# Additional files

## Supplementary files
- MDAR checklist
- Source data 1. Original protein gels, Western blots and autoradiography images.

## Data availability
Cryo-EM maps and atomic coordinates have been deposited with the EMDB and PDB under accession codes EMD-27663 and PDB 8DQX for RFC:PCNA:DNA1 in the open state, codes EMD-27666 and PDB 8DQZ for RFC:PCNA:DNA1 in the intermediate state, codes EMD-27667 and PDB 8DR1 for RFC: PCNA:DNA1 in the closed state, codes EMD-27668 and PDB 8DR2 for RFC:PCNA:DNA2 in the consensus closed state, codes EMD-27669 and PDB 8DR3 for RFC:PCNA:DNA2 in the closed state with NTD, codes EMD-27670 and PDB 8DR4 for RFC:PCNA:DNA2 in the open state without NTD, codes EMD-27671 and PDB 8DR5 for RFC:PCNA:DNA2 in the open state with NTD, codes EMD-27672 and PDB 8DR6 for RFC: PCNA:nDNA in the open state, codes EMD-27673 and PDB 8DR7 for RFC: PCNA:nDNA in the closed state, and codes EMD-27662 and PDB 8DQW for Rad24-RFC:9-1-1:DNA in the open state. PDB 1SJX can be accessed at https://www.rcsb.org/structure/1SXJ, PDB 1T15 can be accessed at https://www.rcsb.org/structure/1T15, PDB 2K6G can be accessed at https://www.rcsb.org/structure/2K6G, PDB 6VVO can be accessed at https://www.rcsb.org/structure/6VVO, PDB 7ST9 can be accessed at https://www.rcsb.org/structure/7ST9, and PDB 7STB can be accessed at https://www.rcsb.org/structure/7STB. Source data are provided with this paper.

The following datasets were generated:

| Author(s) | Year | Dataset title | Dataset URL | Database and Identifier |
|---|---|---|---|---|
| Schrecker M, Castaneda JC, Remus D, Hite RK | 2022 | RFC:PCNA:DNA1 in the open state | https://www.emdataresource.org/EMD-27663 | EMDataResource, EMD-27663 |
| Schrecker M, Castaneda JC, Remus D, Hite RK | 2022 | RFC:PCNA:DNA1 in the open state | https://www.rcsb.org/structure/8DQX | RCSB Protein Data Bank, 8DQX |
| Schrecker M, Castaneda JC, Remus D, Hite RK | 2022 | RFC:PCNA:DNA1 in the intermediate state | https://www.emdataresource.org/EMD-27666 | EMDataResource, EMD-27666 |

*Continued on next page*

*Continued*

| Author(s) | Year | Dataset title | Dataset URL | Database and Identifier |
|---|---|---|---|---|
| Schrecker M, Castaneda JC, Remus D, Hite RK | 2022 | RFC:PCNA:DNA1 in the intermediate state | https://www.rcsb.org/structure/8DQZ | RCSB Protein Data Bank, 8DQZ |
| Schrecker M, Castaneda JC, Remus D, Hite RK | 2022 | RFC:PCNA:DNA1 in the closed state | https://www.emdataresource.org/EMD-27667 | EMDataResource, EMD-27667 |
| Schrecker M, Castaneda JC, Remus D, Hite RK | 2022 | PCNA:DNA1 in the closed state | https://www.rcsb.org/structure/8DR1 | RCSB Protein Data Bank, 8DR1 |
| Schrecker M, Castaneda JC, Remus D, Hite RK | 2022 | RFC:PCNA:DNA2 in the open state without NTD | https://www.emdataresource.org/EMD-27670 | EMDataResource, EMD-27670 |
| Schrecker M, Castaneda JC, Remus D, Hite RK | 2022 | RFC: PCNA:nDNA in the open state | https://www.emdataresource.org/EMD-27672 | EMDataResource, EMD-27672 |
| Schrecker M, Castaneda JC, Remus D, Hite RK | 2022 | RFC:PCNA:DNA2 in the open state with NTD | https://www.rcsb.org/structure/8DR5 | RCSB Protein Data Bank, 8DR5 |
| Schrecker M, Castaneda JC, Remus D, Hite RK | 2022 | RFC:PCNA:DNA2 in the consensus closed state | https://www.emdataresource.org/EMD-27668 | EMDataResource, EMD-27668 |
| Schrecker M, Castaneda JC, Remus D, Hite RK | 2022 | RFC:PCNA:DNA2 in the closed state with NTD | https://www.emdataresource.org/EMD-27669 | EMDataResource, EMD-27669 |
| Schrecker M, Castaneda JC, Remus D, Hite RK | 2022 | RFC:PCNA:DNA2 in the closed state with NTD | https://www.rcsb.org/structure/8DR3 | RCSB Protein Data Bank, 8DR3 |
| Schrecker M, Castaneda JC, Remus D, Hite RK | 2022 | RFC: PCNA:nDNA in the closed state | https://www.emdataresource.org/EMD-27673 | EMDataResource, EMD-27673 |
| Schrecker M, Castaneda JC, Remus D, Hite RK | 2022 | RFC: PCNA:nDNA in the closed state | https://www.rcsb.org/structure/8DR7 | RCSB Protein Data Bank, 8DR7 |
| Schrecker M, Castaneda JC, Remus D, Hite RK | 2022 | RFC:PCNA:DNA2 in the open state with NTD | https://www.emdataresource.org/EMD-27671 | EMDataResource, EMD-27671 |
| Schrecker M, Castaneda JC, Remus D, Hite RK | 2022 | RFC: PCNA:nDNA in the open state | https://www.rcsb.org/structure/8DR6 | RCSB Protein Data Bank, 8DR6 |
| Schrecker M, Castaneda JC, Remus D, Hite RK | 2022 | Rad24-RFC: 9-1- 1: DNA in the open state | https://www.rcsb.org/structure/8DQW | RCSB Protein Data Bank, 8DQW |
| Schrecker M, Castaneda JC, Remus D, Hite RK | 2022 | Rad24-RFC: 9-1- 1: DNA in the open state | https://www.emdataresource.org/EMD-27662 | EMDataResource, EMD-27662 |
| Schrecker M, Castaneda JC, Remus D, Hite RK | 2022 | RFC:PCNA:DNA2 in the consensus closed state | https://www.rcsb.org/structure/8DR2 | RCSB Protein Data Bank, 8DR2 |
| Schrecker M, Castaneda JC, Remus D, Hite RK | 2022 | RFC:PCNA:DNA2 in the open state without NTD | https://www.rcsb.org/structure/8DR4 | RCSB Protein Data Bank, 8DR4 |

The following previously published datasets were used:

| Author(s) | Year | Dataset title | Dataset URL | Database and Identifier |
|---|---|---|---|---|
| Bowman GD, O'Donnell M, Kuriyan J | 2004 | Crystal Structure of the Eukaryotic Clamp Loader (Replication Factor C, RFC) Bound to the DNA Sliding Clamp (Proliferating Cell Nuclear Antigen, PCNA) | https://www.rcsb.org/structure/1SXJ | RCSB Protein Data Bank, 1SXJ |
| Gaubitz C, Liu X, Stone NP, Kelch BA | 2020 | Structure of the human clamp loader (Replication Factor C, RFC) bound to the sliding clamp (Proliferating Cell Nuclear Antigen, PCNA) | https://www.rcsb.org/structure/6VVO | RCSB Protein Data Bank, 6VVO |

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
