## [Editor Report]

The present work uses structural approaches to describe how an ATPase known as a 'clamp loader' opens a ring-shaped clamp protein and binds DNA to promote the deposition of the clamp around a nucleic acid duplex to support chromosomal replication. The paper is important in that it reports new findings on how different regions of the clamp loader bind to and open a clamp, and how the enzyme engages single-stranded and double-stranded regions of target DNAs. Different conformational states of the clamp loader and the clamp are observed, providing a molecular picture of several steps in the clamp loading cycle.

---

## [Decision Letter]

**Decision letter after peer review:**

Thank you for submitting your article "Multistep loading of PCNA onto DNA by RFC" for consideration by *eLife*. Your article has been reviewed by 3 peer reviewers, and the evaluation has been overseen by a Reviewing Editor and Kevin Struhl as the Senior Editor. The reviewers have opted to remain anonymous.

Essential revisions:

1) Some of the conclusions need to toned down a bit, the conclusions regarding the requirement for ATP hydrolysis for ring closure in particular. As pointed out in the reviews, ring closure may occur slowly in the absence of ATP hydrolysis. Just because the PCNA can close on RFC with ATPgS doesn't mean that ATP hydrolysis isn't important for closure. ATP hydrolysis may well speed up clamp loading to a rate needed to keep up with the replication fork.

2) As described below, some revisions should be made to make this work a bit more accessible to a general audience.

3) To rule out the possibility that ATPgS isn not slowly hydrolyzed (or Mg++ ions are dissociating), a better figure for the ATPase sites needs to be included, showing the cryo-EM density for each catalytic center associated with each RFC conformational state.

*Reviewer #1 (Recommendations for the authors):*

1. A central claim of the paper is that nucleotide hydrolysis is not required for clamp closure. Because ATPgS can be hydrolyzed by some proteins, active site density for nucleotide status of *each* RFC subunit needs to be shown for the closed, intermediate and open forms, along with modeling of the nucleotide and Mg++.

2. Even if it turns out that unhydrolyzed nucleotide is indeed present in all active sites throughout all stages, this does not necessarily prove that ATP hydrolysis does not lead to clamp closure under normal operating conditions; e.g., the kinetics of clamp closure may be much faster when hydrolysis can occur. Regardless of the outcome of point 1, the language around this conclusion should be tempered accordingly.

3. The first two sections and first two figures of the Results section describe the RFC-PCNA complex as if DNA isn't present, leading one to think that these perhaps correspond to views of PCNA engagement and opening by RFC prior to DNA binding. However, DNA *is* present and bound to the complex. As a consequence, the implications of the structures as described in the text feel misleading – they really correspond to pre- and post-DNA encirclement states, and as such, they fit into the final model very differently. This distinction needs to be better described and qualified in the text.

4. Lines 170-172. By what evidence are the prior yeast and human RFC-PCNA states really 'inactive', as opposed to corresponding to some kind of pre-clamp opening state? Also, a figure should be included showing how planarity of the RFC ring here differs from one of these states.

5. P. 16. There is a disconnect between the higher PCNA loading activity of RFC-∆N on DNA (as compared to WT RFC) and the higher ATPase activity of WT RFC compared to RFC-∆N. Although this dichotomy is explained away as being due to aggregation of WT RFC on DNA beads, the explanation doesn't fully hold up. The 'dilute' concentrations used for the ATPase assay are the same as those used at the high of the RFC titration in the loading experiments, so why should aggregation be any different? And if aggregation is a problem with the bead assay, then why doesn't it seem to be an issue with the replication assays, where the concentrations of RFC are higher?

6. Figure 6. The mechanistic model implies that the BRCT domain binds DNA on its own, prior to engaging the rest of RFC. This should be tested.

*Reviewer #3 (Recommendations for the authors):*

There are a few points that need some clarification:

1) This study reports the first structure of RFC with full-length Rfc1. It would be helpful particularly to readers outside of the field to add a figure that highlights the different domains of Rfc1 that are discussed throughout the manuscript. Perhaps, show RFC-PCNA with all the subunits but Rfc1 in gray and Rfc1 domains in different colors.

2) A DNA molecule with 20-nt of ds DNA and a 50-base 5' ssDNA overhang was bound to RPA and added to RFC and PCNA for cryo-EM. What happened to the RPA? Was there any density that could be attributed to RPA? Does RPA affect DNA binding or RFC-PCNA conformational states?

3) In terms of the specificity of RFC binding 3' ss/ds DNA junctions, can you clarify how the protein-DNA interactions allow RFC to distinguish between 3' and 5' ss/ds DNA junctions?

4) Figure S3 – It might be helpful to add a key for the residues shown in terms of AAA+ motifs such as Arg finger, sensor 1, etc.

5) The descriptions in the figure legends to Figure S4 and S5 do not match the panels in the figures.

6) When interpreting the data consider the possibility that the populations conformational states of RFC-ATPgammaS complexes and/or the conformations themselves may differ somewhat from those of RFC-ATP complexes because AAA+ proteins are exquisitely sensitive to the bound nucleotide. Some biochemical evidence supports this possibility. Steady-state FRET measurements of PCNA opening showed differences when RFC was bound to ATP vs ATPgammaS (Z. Zhuang et al. PNAS 103, 2546-2551 (2006)). The difference in FRET was interpreted as a smaller opening in PCNA when RFC bound ATPgammaS, but could also be due to a difference in populations of open and closed RFC-PCNA molecules with a larger population of closed RFC-PCNA in measurements with ATPgammaS. Another study showed that RFC could open PCNA when bound by ATPgammaS, but that a smaller fraction of open PCNA molecules formed when RFC bound ATPgammaS than ATP (A. Chiraniya Genes 4, 134-151 (2013)).

---

## [Author Response]

Essential revisions:1) Some of the conclusions need to toned down a bit, the conclusions regarding the requirement for ATP hydrolysis for ring closure in particular. As pointed out in the reviews, ring closure may occur slowly in the absence of ATP hydrolysis. Just because the PCNA can close on RFC with ATPgS doesn't mean that ATP hydrolysis isn't important for closure. ATP hydrolysis may well speed up clamp loading to a rate needed to keep up with the replication fork.

We have revised Figure 6 and added some discussion describing the potential role of ATP hydrolysis in ring closure.

2) As described below, some revisions should be made to make this work a bit more accessible to a general audience.

We have revised the Figures and text as requested.

3) To rule out the possibility that ATPgS isn not slowly hydrolyzed (or Mg++ ions are dissociating), a better figure for the ATPase sites needs to be included, showing the cryo-EM density for each catalytic center associated with each RFC conformational state.

We greatly appreciate the suggestion as it helped to uncover the unique nucleotide specificity of Rfc5. We have added a Figure Supplement showing the density for the bound nucleotides in all states. For Rfc1, Rfc4, Rfc3 and Rfc2, the densities in the high-resolution (>2.4 Å) open and closed states are unambiguously ATPgS. Because the P-S bond is longer than the P-O bond, we can distinguish the two oxygen atoms from the one sulfur atom bound to the γ phosphate. For the Rfc5 subunit, which we had initially modelled as an ADP, we noticed that the density corresponding to nucleotide did not completely agree in high-resolution maps, suggesting that the ligand may have been incorrectly assigned. By comparing the density with other potential nucleotides, we now model the nucleotide as a GDP rather than an ADP. The guanine base is coordinated by the side chain of Rfc5-Arg52, which would be unable to make similarly favorable interactions with an adenine base. As no guanine nucleotides were added to the sample, the GDP is likely to have been co-purified from the yeast. We therefore reasoned that GDP would likely also be present in the Rfc5 subunit of the Rad24-RFC complexes that we recently published (Castaneda et al., 2022). We reprocessed the images and obtained a 2.1 Å reconstruction of Rad24-RFC in complex with 9-1-1. In the improved Rad24-RFC-9-1-1 open state map, we can now unambiguously resolve the ATPgS bound to Rad24, Rfc4, Rfc3 and Rfc2 and GDP bound to Rfc5. On the basis of these structures, we now propose that Rfc5 binds GDP and have revised the text to note the unique nucleotide specificity of Rfc5.

Reviewer #1 (Recommendations for the authors):1. A central claim of the paper is that nucleotide hydrolysis is not required for clamp closure. Because ATPgS can be hydrolyzed by some proteins, active site density for nucleotide status of *each* RFC subunit needs to be shown for the closed, intermediate and open forms, along with modeling of the nucleotide and Mg++.

We agree with the reviewer and greatly appreciate their suggestion. We have added Figure 2—figure supplement 2. showing the density for the bound nucleotides in all states. For Rfc1, Rfc4, Rfc3 and Rfc2, the densities in the high-resolution (>2.4 Å) open and closed states are unambiguously ATPgS. We can resolve two oxygen atoms and one sulfur atom bound to the γ phosphate. For the Rfc5 subunit, which we had initially modelled as an ADP, we noticed that the density corresponding to nucleotide did not completely agree in high-resolution maps, suggesting that the ligand may have been incorrectly assigned. By comparing the density with other potential nucleotides, we now model the nucleotide as a GDP rather than an ADP. The guanine base is coordinated by the side chain of Rfc5-Arg52, which would be unable to make similarly favorable interactions with an adenine base. As no guanine nucleotides were added to the sample, the GDP must have been co-purified from the yeast. We therefore reasoned that GDP would likely also be present in the Rfc5 subunit of the Rad24-RFC complexes that we recently published (Castaneda *et al.*, 2022). We reprocessed the images and obtained a 2.1 Å reconstruction of Rad24-RFC in complex with 9-1-1. In the improved Rad24-RFC-9-1-1 open state map, we can now unambiguously resolve the ATPgS bound to Rad24, Rfc4, Rfc3 and Rfc2 and GDP bound to Rfc5. On the basis of these structures, we now propose that Rfc5 endogenously binds GDP and have revised the text on lines 184-190 to note the unique nucleotide specificity of Rfc5.

2. Even if it turns out that unhydrolyzed nucleotide is indeed present in all active sites throughout all stages, this does not necessarily prove that ATP hydrolysis does not lead to clamp closure under normal operating conditions; e.g., the kinetics of clamp closure may be much faster when hydrolysis can occur. Regardless of the outcome of point 1, the language around this conclusion should be tempered accordingly.

We agree with the reviewer and raise this possibility in the discussion (lines 478 – 486) along with relevant references. We have revised Figure 6 to highlight the possible role of ATP hydrolysis in clamp closure.

3. The first two sections and first two figures of the Results section describe the RFC-PCNA complex as if DNA isn't present, leading one to think that these perhaps correspond to views of PCNA engagement and opening by RFC prior to DNA binding. However, DNA *is* present and bound to the complex. As a consequence, the implications of the structures as described in the text feel misleading – they really correspond to pre- and post-DNA encirclement states, and as such, they fit into the final model very differently. This distinction needs to be better described and qualified in the text.

We agree with the reviewer that the correspondence between PCNA conformation and DNA state was not clear in the initial submission. As the reviewer states, the open conformation of the complex in the presence of DNA1 presented in Figures 1 and 2 corresponds to a pre-DNA encirclement state. However, we do resolve structures in the presence of DNA2 or nicked DNA in which the DNA is located in site 1 (central chamber), yet the PCNA ring remains open (Figure 3—figure supplement 3 and Figure 3—figure supplement 6). In addition, recent work from Brian Kelch’s group (as mentioned by Reviewer 2 below) showed that ring can adopt an ensemble of open and closed configurations in the absence of DNA (Gaubitz et al., 2022b). Taken together, these data indicate that PCNA can adopt an ensemble of conformations when bound to RFC.

To clarify that the complexes presented in Figures 1 and 2 were determined in the presence of DNA, we have added the DNA to Figure 1C-E and noted in the figure legend for Figure 2 that DNA is removed for clarity. We have also clarified the text on lines 582-584 to note that PCNA can adopt an ensemble of conformations, including the open, intermediate, and closed states when bound to RFC in the presence of ATPgS and that while DNA substrates binding at sites 1, 2 or 3 may alter the equilibrium between these states, it is not required for PCNA opening.

4. Lines 170-172. By what evidence are the prior yeast and human RFC-PCNA states really 'inactive', as opposed to corresponding to some kind of pre-clamp opening state? Also, a figure should be included showing how planarity of the RFC ring here differs from one of these states.

The ’inactive’ state, which has also been referred to as ‘autoinhibited’ in the literature (Gaubitz et al., 2020), refers to the catalytically incompetent active site conformations observed in previous structures of yeast and human RFC:PCNA complexes off DNA. We agree with the reviewer that this state may very well be on the pathway to clamp loading, preceding clamp opening. We now clarify this point in the introduction (lines 78-80) where we first introduce this state and use the term autoinhibited in the text in figures for consistency.

5. P. 16. There is a disconnect between the higher PCNA loading activity of RFC-∆N on DNA (as compared to WT RFC) and the higher ATPase activity of WT RFC compared to RFC-∆N. Although this dichotomy is explained away as being due to aggregation of WT RFC on DNA beads, the explanation doesn't fully hold up. The 'dilute' concentrations used for the ATPase assay are the same as those used at the high of the RFC titration in the loading experiments, so why should aggregation be any different? And if aggregation is a problem with the bead assay, then why doesn't it seem to be an issue with the replication assays, where the concentrations of RFC are higher?

The PCNA loading assay utilizes DNA immobilized on paramagnetic beads, whereas ATPase and replication assays utilize DNA free in solution. We have noticed that RFC has a tendency to stick non-specifically to the beads, which we hypothesize may explain the disconnect between the PCNA loading and ATPase data for full-length RFC.

Moreover, in preliminary ATPase assays, we noticed that the degree of stimulation of RFC activity by both PCNA and DNA is inversely correlated with RFC concentration. Based on the fact that previous studies had noted stability issues of RFC (e.g. on page 17 we reference a study by the Burgers laboratory [Gomes et al., JBC, 2000] to highlight this fact), we reasoned that this observation may be due to the potential aggregation of RFC at higher concentrations. Consistent with this notion, we observe a reduction in replication activity at increased RFC concentration (e.g. compare lanes 4 + 5 in Figures 5 C+E).

Alternatively, the reduced PCNA loading activity of full-length RFC relative to RFC-1DN may be due to the increased non-specific DNA binding activity of full-length RFC, as has been suggested before. We now include the following statement in the Results section (page 17, lines 394-397) to acknowledge this possibility: “Alternatively, the reduced PCNA loading activity of full-length RFC relative to RFC-1DN observed in this assay may be due to the previously noted non-specific dsDNA binding activity of full-length RFC, which may limit RFC turnover at 3’ ss/dsDNA junctions (Gomes et al., 2000; Podust et al., 1998; Uhlmann et al., 1997)”.

6. Figure 6. The mechanistic model implies that the BRCT domain binds DNA on its own, prior to engaging the rest of RFC. This should be tested.

Previous studies with the isolated RFC1 BRCT domain of various organisms have demonstrated that this domain binds DNA on its own, supporting our model. To highlight this fact, we now include the following statement in the Discussion (page 22, line 507): “Previous studies have demonstrated that the BRCT domain of RFC1 forms an autonomous dsDNA binding domain (Allen et al., 1998; Burbelo et al., 1993; Fotedar et al., 1996; Kobayashi et al., 2006).”

Reviewer #3 (Recommendations for the authors):There are a few points that need some clarification:1) This study reports the first structure of RFC with full-length Rfc1. It would be helpful particularly to readers outside of the field to add a figure that highlights the different domains of Rfc1 that are discussed throughout the manuscript. Perhaps, show RFC-PCNA with all the subunits but Rfc1 in gray and Rfc1 domains in different colors.

As requested, we have added two additional panels to Figure 1 to show the domain arrangement of the subunits and the 3D architecture of Rfc1 alone.

2) A DNA molecule with 20-nt of ds DNA and a 50-base 5' ssDNA overhang was bound to RPA and added to RFC and PCNA for cryo-EM. What happened to the RPA? Was there any density that could be attributed to RPA? Does RPA affect DNA binding or RFC-PCNA conformational states?

As mentioned in lines 138-139, we have included RPA in the reactions because RPA was previously found to promote the loading of PCNA by RFC specifically at 3’ ss/dsDNA junctions. However, we do not observe cryo-EM density corresponding to RPA in our maps. To clarify this point, we now include the following statement in the Results section (lines 150-152): “Despite being present in the vitrified sample, we did not detect unassigned densities potentially corresponding to RPA in the electron density maps.”

3) In terms of the specificity of RFC binding 3' ss/ds DNA junctions, can you clarify how the protein-DNA interactions allow RFC to distinguish between 3' and 5' ss/ds DNA junctions?

Our data are not conclusive on the mechanisms by which 3’ ss/dd DNA junctions are recognized by RFC. The DNA is partially melted in all of the structures that we were able to resolve. Future work identifying the initial association state or states will be necessary to provide a better understanding of how RFC can distinguish between 3’ and 5’ ss/dd DNA junctions.

4) Figure S3 – It might be helpful to add a key for the residues shown in terms of AAA+ motifs such as Arg finger, sensor 1, etc.

We have added a sequence alignment with the key residues highlighted.

5) The descriptions in the figure legends to Figure S4 and S5 do not match the panels in the figures.

Corrected.

6) When interpreting the data consider the possibility that the populations conformational states of RFC-ATPgammaS complexes and/or the conformations themselves may differ somewhat from those of RFC-ATP complexes because AAA+ proteins are exquisitely sensitive to the bound nucleotide. Some biochemical evidence supports this possibility. Steady-state FRET measurements of PCNA opening showed differences when RFC was bound to ATP vs ATPgammaS (Z. Zhuang et al. PNAS 103, 2546-2551 (2006)). The difference in FRET was interpreted as a smaller opening in PCNA when RFC bound ATPgammaS, but could also be due to a difference in populations of open and closed RFC-PCNA molecules with a larger population of closed RFC-PCNA in measurements with ATPgammaS. Another study showed that RFC could open PCNA when bound by ATPgammaS, but that a smaller fraction of open PCNA molecules formed when RFC bound ATPgammaS than ATP (A. Chiraniya Genes 4, 134-151 (2013)).

We thank that reviewer for highlighting the differences between ATP and ATPgS in the conformational landscape of states adopted by RFC. We have now added a section to the discussion describing the potential differences.